# Structural bases of inhibitory mechanism of Ca$_V$1.2 channel inhibitors

Yiqing Wei [1,2,6], Zhuoya Yu [1,2,6], Lili Wang [3,6], Xiaojing Li [1,2,6], Na Li [4], Qinru Bai [1,2], Yuhang Wang [1,2], Renjie Li [1,2], Yufei Meng [1,2], Hao Xu [5], Xianping Wang [1], Yanli Dong [1], Zhuo Huang [3] ✉, Xuejun Cai Zhang [1,2] ✉ & Yan Zhao [1,2] ✉

The voltage-gated calcium channel Ca$_V$1.2 is essential for cardiac and vessel smooth muscle contractility and brain function. Accumulating evidence demonstrates that malfunctions of Ca$_V$1.2 are involved in brain and heart diseases. Pharmacological inhibition of Ca$_V$1.2 is therefore of therapeutic value. Here, we report cryo-EM structures of Ca$_V$1.2 in the absence or presence of the antirheumatic drug tetrandrine or antihypertensive drug benidipine. Tetrandrine acts as a pore blocker in a pocket composed of S6$^{II}$, S6$^{III}$, and S6$^{IV}$ helices and forms extensive hydrophobic interactions with Ca$_V$1.2. Our structure elucidates that benidipine is located in the D$_{III}$-D$_{IV}$ fenestration site. Its hydrophobic sidechain, phenylpiperidine, is positioned at the exterior of the pore domain and cradled within a hydrophobic pocket formed by S5$^{DIII}$, S6$^{DIII}$, and S6$^{DIV}$ helices, providing additional interactions to exert inhibitory effects on both L-type and T-type voltage gated calcium channels. These findings provide the structural foundation for the rational design and optimization of therapeutic inhibitors of voltage-gated calcium channels.

Voltage-gated calcium channels can be classified into high voltage activated calcium channels (HVAs) and low voltage activated calcium channels (LVAs)[1,2]. HVA (Ca$_V$1 and Ca$_V$2) contains pore-forming subunit α1, auxiliary subunits α2δ and β, whereas LVA (Ca$_V$3) contains only α1. Ca$_V$1 channels are defined as L-type due to their long-lasting current[3]. L-type Ca$_V$1.2 plays an essential role in cardiac and smooth muscle contractility, neuroendocrine regulation, and multiple other processes[4–6]. It is expressed in the brain, cardiomyocytes, sinoatrial node, vascular system, pancreatic islets, arsenal medulla, and intestinal/bladder smooth muscle. Ca$_V$1.2 is predominant in contraction in atrial, ventricular, and vascular excitable tissues as it triggers the release of calcium from the sarcoplasmic reticulum terminal cistern for actin-myosin interaction and thus is the uppermost target for the treatment of hypertension and one of the therapeutic targets of arrhythmia[4]. Meanwhile, approximately 90% of the L-type calcium channels in the brain are Ca$_V$1.2 (refs. 7–9). It is highly expressed in hippocampal neurons, participates in long-term synaptic plasticity and is essential for the formation of spatial memory, fear memory, and emotional behaviors[9–14]. Its malfunction is related to schizophrenia and bipolar affective disorder[15]. Ca$_V$1.2 also plays an essential role in early development, regulating chondrogenesis[16,17]. Inherited mutations of Ca$_V$1.2 are associated with serious diseases, including cardiac and vessel diseases as well as neuronal disorders[18–20].

The discovery of more drug binding patterns is one of the practical goals in structural biology. There are various marketed drugs targeting Ca$_V$1.2. In previous studies, the structures of L-type calcium channels bound with dihydropyridine (commercial name: nifedipine), phenylamine (verapamil), benzothiazine (diltiazem), and

[1]Key Laboratory of Biomacromolecules (CAS), National Laboratory of Biomacromolecules, CAS Center for Excellence in Biomacromolecules, Institute of Biophysics, Chinese Academy of Sciences, Beijing 100101, China. [2]College of Life Sciences, University of Chinese Academy of Sciences, Beijing 100049, China. [3]State Key Laboratory of Natural and Biomimetic Drugs, Department of Molecular and Cellular Pharmacology, School of Pharmaceutical Sciences, Peking University Health Science Center, Beijing 100191, China. [4]Heart Center and Beijing Key Laboratory of Hypertension, Beijing Chaoyang Hospital, Capital Medical University, Beijing 100020, China. [5]Division of Life Sciences and Medicine, University of Science and Technology of China, Hefei 230026, China. [6]These authors contributed equally: Yiqing Wei, Zhuoya Yu, Lili Wang, Xiaojing Li. ✉e-mail: huangz@hsc.pku.edu.cn; zhangc@ibp.ac.cn; zhaoy@ibp.ac.cn

diphenylmethyl piperazine (cinnarizine) were resolved[21,22]. However, there are still unidentified inhibition mechanisms of clinical drugs remaining to be clarified. For instance, tetrandrine is a dibenzyl iso-quinoline alkaloid and functions as a voltage gated calcium channel antagonist[23–25]. Similar to the well-known artemisinin, tetrandrine is an effective ingredient extracted from traditional Chinese medicine and is attracting more attention due to its diverse pharmacological effects. It is currently used in the clinic for anti-rheumatism and analgesia, as well as for the treatment of lung cancer and silicosis, and shows an effect of lowering blood pressure[26–30]. Analyzing the binding mode between $Ca_V1.2$ and tetrandrine can expand our understanding of ion channel inhibitors and provide structural bases for further modification and optimization of such drugs. However, different sidechain modifications of dihydropyridine may cause different pharmacological effects and varied affinity toward ion channels. For example, in guinea pig ventricular cells, the $IC_{50}$ values of benidipine and nifedipine are at least 10 times different, providing a reference for affinity optimization of dihydropyridine or other drugs. To explain these phenomena, further analyses of more dihydropyridine drugs are still desirable. Moreover, many pathogenic mutations cause distortion of the electrophysiological properties of $Ca_V1.2$. As pathogenic mutation sites are usually not strictly conserved in L-type calcium channels, the lack of $Ca_V1.2$ structure limits the understanding of the pathogenic mechanism of these mutations.

In this study, we determine the near-atomic resolution cryo-EM structure of apo $Ca_V1.2$ and analyzed the distribution pattern as well as potential impacts on the gating of a collection of pathogenic mutations in $Ca_V1.2$. In addition, we solve the $Ca_V1.2$ complex structures in the presence of tetrandrine or benidipine. Tetrandrine blocks the pore of $Ca_V1.2$, interacting with hydrophobic residues in the central cavity. The mutations N741W (Asn-to-Trp mutation at the position 741) and N1179W can significantly reduce the inhibition of tetrandrine, confirming that a large cavity space is needed for tetrandrine blocking. Benidipine is located in the fenestration between $D_{III}$ and $D_{IV}$. The hydrophobic sidechain of benidipine provides extra interactions with the channel subunit α1, supporting the inhibition of both L- and T-type voltage gated calcium channels. These results elucidate how the chemical groups of these inhibitors subtly remodeled their interactions with $Ca_V1.2$.

## Results and discussion

### Architecture of the $Ca_V1.2$ channel complex

To investigate the inhibitory mechanism of $Ca_V1.2$-targeting drugs, we co-expressed the human $Ca_V1.2$ with its physiological auxiliary subunits α2δ1 and β2b in HEK293 cells. The genes encoding these three subunits were subcloned into pEG-BacMam vector. We also carried out whole-cell patch clamp using HEK293T cells to characterize the functional properties of these constructs (Supplementary Fig. 1a). The subunits expressed with these constructs form a functional complex, showing an activation curve similar to those from previous studies[31,32], and the half-activation voltage ($V_{1/2}$) of the recombinant channel is +1.0 (±0.5) mV. Subsequent purification experiments resulted in a sharp and symmetrical SEC profile, indicating homogeneity of the sample of the channel complex. The SDS-PAGE gel confirmed that the purified sample contained all three subunits (Supplementary Fig. 1b, c). Next, we collected cryo-EM data using Titan2 Krios and carried out single-particle cryo-EM analysis. The three-dimensional cryo-EM maps of $Ca_V1.2$ in the apo state (denoted as $Ca_V1.2^{apo}$), tetrandrine bound state ($Ca_V1.2^{TET}$), and benidipine bound state ($Ca_V1.2^{BEN}$) were determined at 3.5-Å, 3.4-Å, and 3.3-Å resolutions, respectively. These maps are rich in structural features, including densities for sidechains, N-glycans, and lipid molecules, and allowed us to reliably build atomic models of these $Ca_V1.2$ complexes (Supplementary Figs. 2–3) (Supplementary Table 1).

The structure of the $Ca_V1.2$ channel complex is composed of a pore-forming α1 subunit and auxiliary subunits α2δ1 and β2b (Fig. 1a, b). Its α1 subunit consists of four repeat transmembrane domains ($D_I$–$D_{IV}$). Each domain contains six transmembrane helices (S1–S6) and forms the channel in a domain-swapped fashion. The S1–S4 helices constitute the voltage sensing domain (VSD) to detect changes in the cross-membrane electrostatic potential. Consistent with all known structures of voltage-gated channels, the S4 helix assumes a $3_{10}$-helical conformation. The S5–S6 regions of all four domains form the pore domain to facilitate and control the passage of ions (Fig. 1c, d). The selectivity filter (SF) is contributed by the re-entrant loops from each subunit that connect S5 and S6 and include P1 and P2 helices (Fig. 1e). In contrast to the conformational heterogeneity of the α-interaction domain (AID) found in $Ca_V1.1$ (ref. 33), AID is well resolved in the current $Ca_V1.2$ structure, with its N- and C-termini adjacent to the intracellular gate and $VSD_{II}$, respectively (Fig. 1a). The selectivity filter contains four acidic residues ($E363^{DI}$, $E706^{DII}$, $E1135^{DIII}$, and $E1464^{DIV}$; i.e. the EEEE motif), one from each domain, producing a strongly negatively charged zone to attract cations and determine selectivity for $Ca^{2+}$ ions[33] (Fig. 1e). On the cytoplasmic side, the four S6 helices bundle together and act as a gate to control the access of ions into the pore (Fig. 1f). The pore diameter profile indicates that the current structure represents an inactivated state with a closed intracellular gate (Fig. 1d). Although all S4 helices in the four VSDs are in an 'up' conformation, the intracellular gate remains closed, suggesting that the structure of $Ca_V1.2$ was captured in an inactivated state. In addition, the α2δ1 subunit was clearly resolved and sits on the extracellular side of $Ca_V1.2$, interacting with the E149, D150, and D151 residues from the $D_I$ repeat. The β2b subunit is composed of a GK domain and an SH3 domain and is associated with the cytosolic AID region of $Ca_V1.2$ (Fig. 1). A lipid molecule was observed in the $D_I$–$D_{II}$ fenestration of $Ca_V1.2^{apo}$, consistent with other structure studies on $Ca_V1.2$ (refs. 34–36).

The structure of $Ca_V1.2^{apo}$ was compared with that of $Ca_V1.1$ (PDB ID: 5GJW)[33], giving rise to an RMSD of 1.35 Å for 1772 $C_α$-pairs. The overall structure, including VSD, SF, and extracellular loops (ECLs), is fairly superimposable (Supplementary Fig. 4a). In both structures, in the absence of applied membrane potential, all VSDs show an "up" conformation of the S4 helices and are thus in the "activated" state. However, some structural differences are observed between the structures of $Ca_V1.2^{apo}$ and $Ca_V1.1$. For example, two π-bulges occur on the $S6^{DI}$ and $S6^{DIII}$ helices of $Ca_V1.2^{apo}$, whereas both helices assume a canonical α-helix conformation in $Ca_V1.1$. The sidechains of residues on the intracellular segment, starting from $F394^{S6I}$ and $F1175^{S6III}$, undergo an approximate 90° rotation (Supplementary Fig. 4b). The residues forming the central cavity and the intracellular gate differs from that in $Ca_V1.1$. In specific, the side chain of F394 and F1175 face the central cavity, causing a shrinkage of the inner radius of the pore domain (Supplementary Fig. 4c, d). In $Ca_V1.1$, the intracellular gate is constituted by V329, L333, F656, A660, F1060, V1064 F1376, and I1380. With the local conformational shifts on the S6 helices in $Ca_V1.2$, the intracellular gate is formed by L401, S405, L749, V753, V1182, I1186, F1519, and I1523 (Supplementary Fig. 4e). A previous study on TRPV6 channel suggested that the transition from an α-helix to a π-helix may prompt channel opening[37]. However, despite such transition being observed in $Ca_V1.2$, its intracellular gate remains closed (Supplementary Fig. 4f).

### Pathogenic mutations of $Ca_V1.2$

Dysfunction of $Ca_V1.2$ usually causes heart and vessel diseases and neurological disorders[18–20]. At least 86 mutations of the $Ca_V1.2$ α1 subunit have been identified in various cases of human diseases, including Timothy syndrome (TS), long QT syndrome 8 (LQTS), short QT syndrome (SQTS), Brugada syndrome (BrS), autism, and schizophrenia, among others (Supplementary data). Our current

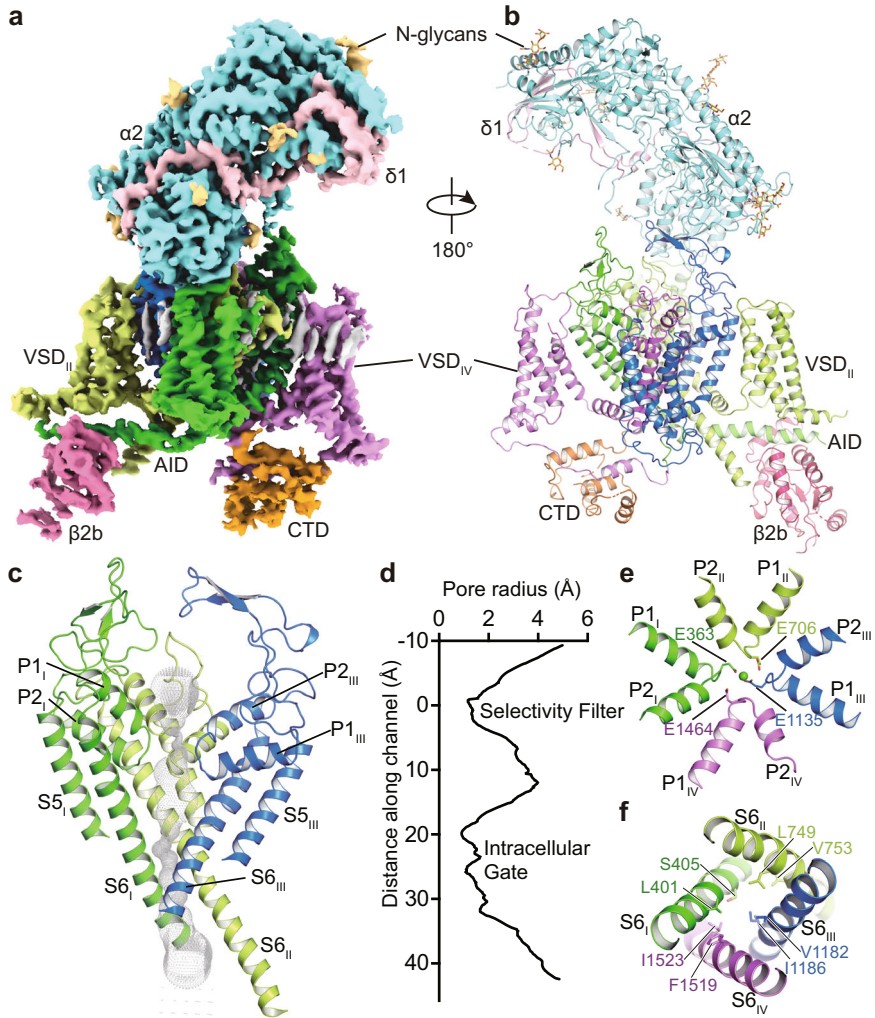

**Fig. 1 | Architecture of the Ca_V1.2 complex. a, b** The overall EM density map and structure of the Ca_V1.2 complex. The α2δ1 and β2b subunits, voltage-sensing domain (VSD), C-terminal domain (CTD), α-interacting domain (AID), and N-glycans were labeled. The α1 subunit is colored in deep green (Domain I, D_I: A114-R211, F231-I447), light green (D_II: W509-K778), deep blue (D_III: F889-I943, F964-Y1197), and mauve (D_IV: K1198-S1321, E1350-T1357, S1372-E1484, P1494-D1533), and orange (CTD: W1534-P1574, C1582-S1593, N1607-K1638, K1647-P1655), respectively. The β2b (D222-R276, L297-W407), α2 (F27-D129, Q143-V224, K232-N540, Q554-K690, N698-T828, D849-P906), and δ1 (Q972-G1022, T1028-D1070) subunits are colored in magenta, turquoise, and pink, respectively. N-glycans are colored in yellow. **c** The ion permeation path in the pore domain. The selectivity filter and S5-S6 helices are shown in cartoon and viewed in parallel to the membrane plane. The ion conducting pathway was calculated by using the program HOLE and illustrated with gray dots. **d** Plot of pore radii for the Ca_V1.2 complex. **e** The selectivity filter ring of four glutamate residues from the four domains of Ca_V channel were shown in sticks. A cation ion is shown as a green sphere. **f** The intracellular gate formed by four S6 helices viewed from the intracellular side. Hydrophobic residues are shown in sticks.

Ca_V1.2 structure provides a template to map these disease-related mutations into the 3D structure. Among these mutants, 39 sites are distributed on the four VSDs, pore domain, and C-terminal domain (CTD) (Fig. 2). The VSD_II and pore domain harbor most of the pathogenic mutations. The remaining 47 mutation sites are mainly distributed on the N-terminus, cytosolic long linkers between different domains (mainly D_II–D_III), and the C-terminal tail, and they were not determined due to conformational heterogeneity.

Three gain-of-function (GOF) mutations have been identified and verified by electrophysiological experiments. G402S and G406R were found in Timothy syndrome cases, and they inhibit voltage-dependent inactivation of Ca_V1.2 (refs. 18,38). Since all G406 substitutions to other amino acid residues, such as alanine, serine, and valine, impair the inactivation process, the function of G406 is proposed as irreplaceable[38]. In our Ca_V1.2 structure, both G402 and G406 are located on the S6^DI helix and close to the intracellular gate (Supplementary Fig. 5b). We speculate that the absence of a sidechain in glycine at these two positions is essential for the conformational

transition from the open state to the inactivated state, and replacement by a sidechain-containing residue at either of these two positions probably hinders the transition.

In a previous study, L762F associated with LQTS was found to result in a GOF defect with slower inactivation[39]. In the current Ca_V1.2 structure, L762 on the S6^DII helix is located around the cytosolic membrane surface. Its sidechain points toward the AID motif and the S4-S5^DII linker helix (Supplementary Fig. 5c). We speculate that the mutation L762F results in steric hindrance with AID or S4-S5^DII helices, leading to a conformational change in these helices and affecting channel inactivation kinetics.

N300D is a loss-of-function (LOF) mutation and has been identified in Brugada syndrome patients[40]. This mutation leads to a decrease in the current density by impairing the expression level of Ca_V1.2 in the membrane[40]. N300 is located in the extracellular loop between S5^DI and S6^DI (Supplementary Fig. 5a) and is exposed to the interface between the α1 and α2δ1 subunits, which is essential for trafficking of the Ca_V1.2 complex[41,42]. N300D may thus affect traffic by disturbing the

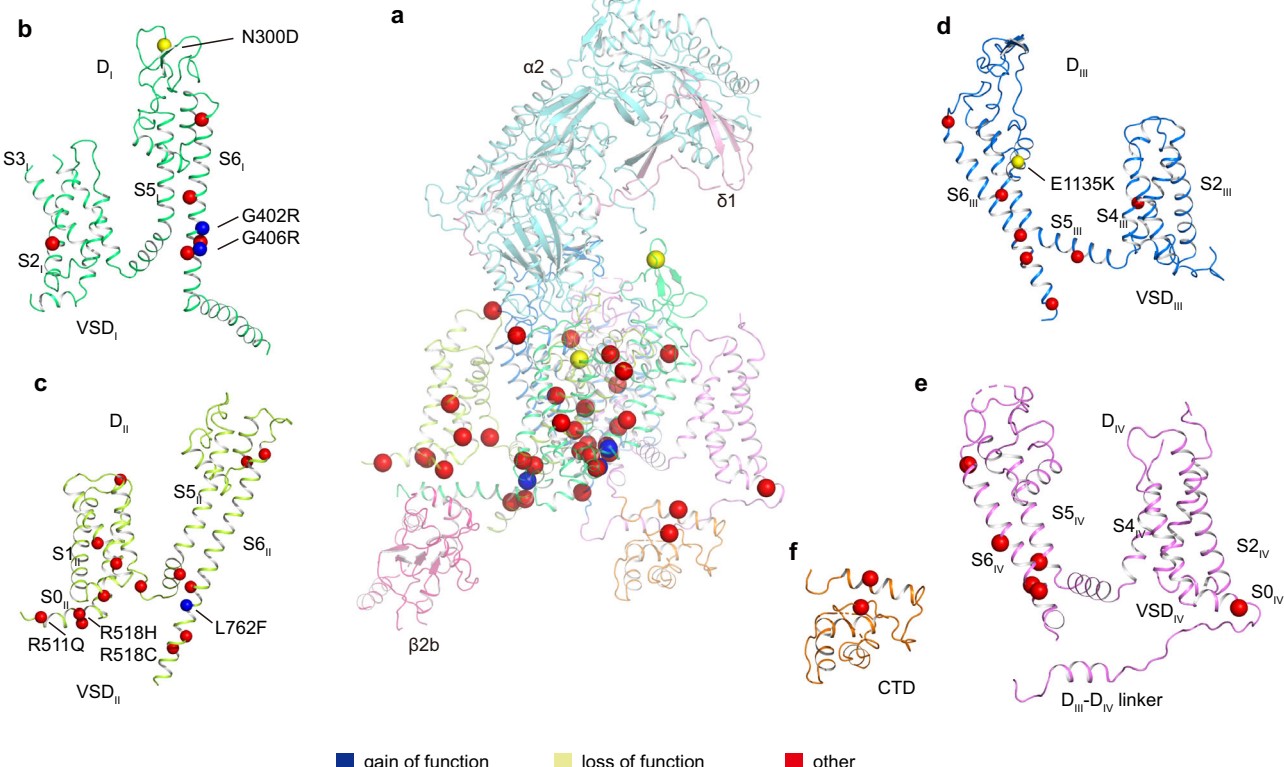

**Fig. 2 | Molecular mapping of pathogenic mutations. a–f** Disease related mutations are located on $D_I$ (**b**), $D_{II}$ (**c**), $D_{III}$ (**d**), $D_{IV}$ (**e**), and CTD (**f**). $D_I$, $D_{II}$, $D_{III}$, $D_{IV}$, and CTD of α1 subunit are colored in deep green, light green, deep blue, mauve, and orange, respectively. Subunits α2, δ1, and β2b are colored in turquoise, pink, and magenta, respectively. Mutation sites are marked with spheres and colored in blue (gain of function), yellow (loss of function), and red (other), respectively.

interaction with α2δ1. In addition, E1135K is another LOF mutation identified in patients with idiopathic QT prolongation, bradycardia, and autism spectrum disorder[43,44]. This mutation is positioned in the EEEE motif of the selectivity filter in the current $Ca_V1.2$ structure (Fig. 1e and Supplementary Fig. 5d). Electrophysiological experiments demonstrated that the E1135K mutation makes $Ca_V1.2$ a nonselective monovalent cation channel, in line with functional roles of the EEEE motif in determining ion selectivity[44,45].

Previous studies have indicated that R511Q, R518C, and R518H are associated with LQTS and cardiac only Timothy syndrome[46,47]. In the current $Ca_V1.2$ structure, R511 and R518 are located at a cytosolic amphiphilic helix N-terminal to S1$^{DII}$ (denoted as S0$^{DII}$). The S0$^{DII}$ helix is rich in positive charged residues ($^{511}$RFCRRKCRAAVK$^{522}$) that electrostatically interact with the negatively charged C-terminus of AID (Supplementary Fig. 5e). Electrophysiological experiments demonstrated that pathogenic mutations associated with R511 and R518 result in slower inactivation[46,47]. We speculate that these mutations interfere with the interaction between S0$^{DII}$ and AID and thus affect the conformation of AID relative to the pore domain, resulting in reduced inactivation. This speculation is consistent with the notion that AID is important for regulating channel inactivation[48,49].

### Recognition of tetrandrine

Tetrandrine is a traditional Chinese clinical agent for autoimmune disorders, cardiovascular diseases, and hypertension. It is extracted from the root of *Stephania tetrandra* S Moore and inhibits both L-type and T-type $Ca_V$ channels[25,50,51]. To explore the molecular basis how tetrandrine blocks the activity of $Ca_V1.2$, tetrandrine (10 μM) was added during protein expression and purification. We determined the tetrandrine-bound $Ca_V1.2$ complex ($Ca_V1.2^{TET}$) at 3.4-Å resolution (Supplementary Fig. 2). In our cryo-EM map, we found a well-resolved triangle-shaped density in the central cavity of the pore domain,

which is well fitted with the tetrandrine molecule and positioned proximal to the selectivity filter (Fig. 3a–c). The two major moieties of the 6,7-dimethoxy-2-methyl-1-benzyl-isoquinoline ring are positioned close to the $D_{II}$–$D_{III}$ fenestration site and $D_{IV}$, and they are stabilized by forming hydrophobic or Van Der Waals interactions with surrounding residues from the selectivity filter and S6 helices (Fig. 3d, e). In particular, the residues F1175$^{S6III}$, M1178$^{S6III}$, A1512$^{S6IV}$, I1516$^{S6IV}$, and F1519$^{S6IV}$ play important roles in stabilizing tetrandrine. The 6,7-dimethoxy and 2/2′-N groups from the isoquinoline ring form hydrogen bonds with the N741$^{S6II}$ and Y1508$^{S6IV}$/N1179$^{S6III}$ residues. In addition, in agreement with observations in drug-bound $Ca_V3.3$ complex structures[52], a lipid molecule penetrates through the fenestration site between $D_I$ and $D_{II}$ and participates in hydrophobic interactions with the tetrandrine molecule (Fig. 3d, e). Compared with other drug binding modes observed in the previously reported $Ca_V1.1$ structures, we found that the tetrandrine binding mode is distinct from those of nifedipine (a DHP derivative), verapamil (PAA), diltiazem (BTZ), and cinnarizine (CIN) (Supplementary Fig. 6a). In particular, tetrandrine completely squeezes the binding site of BTZ (Supplementary Fig. 6b), consistent with previous experimental results showing that tetrandrine and diltiazem cannot simultaneously bind to L-type $Ca_V$ channels[53]. To validate the binding site of tetrandrine, we designed two mutants, N741W and N1179W, with the potential to create steric clash between tetrandrine and the binding site. We determined the dose-response curves of tetrandrine for both wild-type and these two mutants. The curves for both mutants exhibited a rightward shift compared to that of wild-type $Ca_V1.2$, indicating that these mutants reduce the sensitivity of these mutants to tetrandrine. The derived $IC_{50}$ values for tetrandrine with the N741W and N1179W mutants are 71.3 nM and 41.8 nM, respectively, which are higher than that of wild-type (~14.2 nM) (Fig. 3), further confirming the binding site of tetrandrine.

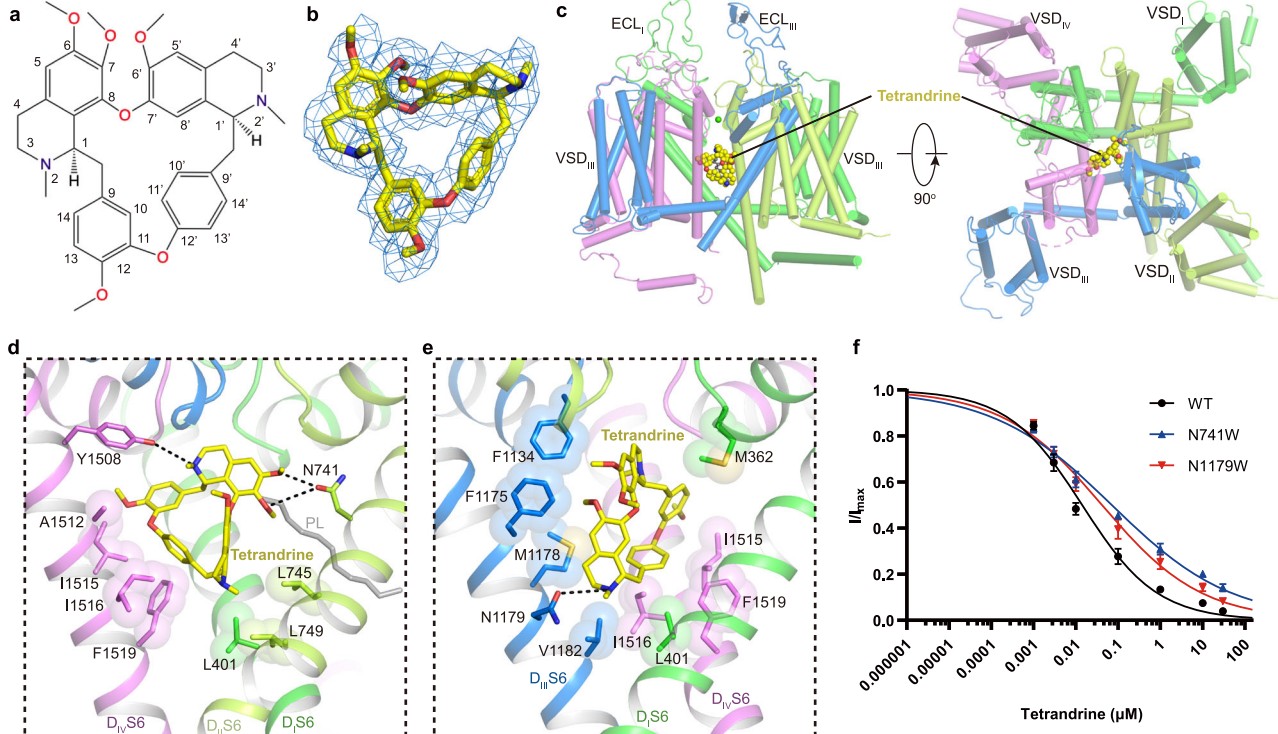

**Fig. 3 | Structure basis for blockade of Ca$_V$1.2 by tetrandrine. a** Chemical structure of tetrandrine. **b** The cryo-EM density shown in blue mesh for tetrandrine in sticks. **c** The overall structure of the Ca$_V$1.2$^{TET}$ complex. The domains of Ca$_V$1.2$^{TET}$ are colored as D$_I$ in deep green, D$_{II}$ in light green, D$_{III}$ in deep blue, and D$_{IV}$ in mauve. The tetrandrine located in the pore domain is presented as yellow spheres. **d**, **e** Detailed binding sites for tetrandrine showing interactions between tetrandrine and Ca$_V$1.2. The sidechains of key residues are displayed in sticks and the hydrophobic side chains are overlaid with transparent surfaces. Black dashed lines indicate potential hydrogen bonds. A lipid located within the fenestration site is shown in gray. **f** Dose-response curves of tetrandrine inhibition on Ca$_V$1.2$^{WT}$, Ca$_V$1.2$^{N741W}$, and Ca$_V$1.2$^{N1179W}$. Ca$_V$1.2$^{N741W}$ and Ca$_V$1.2$^{N1179W}$ shift the IC$_{50}$ of tetrandrine on Ca$_V$1.2 from 14.2 nM to 71.3 nM and 41.8 nM, respectively. Data in curves are represented as mean ± SEM (*n* = 10 biologically independent experiments). Source data are provided as a Source Data file.

## Antagonism of benidipine

1,4-Dihydropyridine derivatives (DHPs), such as nifedipine, selectively inhibit certain voltage gated calcium channels and are widely used clinically for the treatment of hypertension[54]. However, DHPs frequently cause side effects such as proteinuria, upon chronic use. The specific inhibition of L-type calcium channels is usually an advantage but is a defect in the kidney. The reason is that L-type channels are distributed in the kidney glomerulus afferent arteriole but are absent in the efferent arteriole. This inconsistent inhibition results in unbalanced local blood pressure, leading to kidney damage with proteinuria[55,56]. However, benidipine, another DHP derivative, can additionally inhibit T-type calcium channels, which are more abundant in efferent arterioles than in afferent arterioles, making blood pressure more balanced in different parts of the kidney. Compared with nifedipine and amlodipine, benidipine causes fewer adverse side effects, such as proteinuria and leg edema[55,57,58]; therefore, it is widely used in the clinic. Compared with nifedipine, benidipine replaces 8-nitrophenyl with 9-nitrophenyl and 3-carboxylic acid methyl ester with 3-(1-phenylmethyl-3-piperidinyl) ester. By supplementing benidipine when solubilizing the membrane and before preparing the cryo-EM sample, we obtained a 3.3-Å resolution benidipine-bound Ca$_V$1.2 structure (Ca$_V$1.2$^{BEN}$) (Supplementary Fig. 2).

The benidipine molecule penetrates into the D$_{III}$-D$_{IV}$ fenestration and corresponds to a 'claw' shape density in the map (Fig. 4a, b). Unlike the pore-blocker tetrandrine, benidipine does not directly block the pore but allosterically inhibits Ca$_V$1.2. The main moiety of dihydropyridine is biased towards D$_{III}$ (Fig. 4c). The nitrogen atom on the pyridine ring forms a potential hydrogen bond with S1132$^{P1}$ (Fig. 4d). The nitrogen atom on 9-nitrophenyl and the oxygen atom on 5-dicarboxylic acid of the pyridine ring form hydrogen bonds with

T1056$^{S5III}$ and Q1060$^{S5III}$, respectively (Fig. 4d). Whereas similar interactions with T935 and Q939 have been identified in the structure of the Ca$_V$1.1 complex, benidipine also contains a phenylmethyl piperidinyl group, which is positioned in a hydrophobic pocket formed by residues I1046$^{S5III}$, I1049$^{S5III}$, V1053$^{S5III}$, F1181$^{S6III}$, M1509$^{S6IV}$, and F1513$^{S6IV}$, providing additional hydrophobic interactions to stabilize its binding, in line with the inhibitory IC$_{50}$ of benidipine being approximately ten times smaller than that of nifedipine[59] (Fig. 4d, e). Previous studies indicated that nifedipine specifically inhibits L-type Ca$_V$ channels, showing no sensitivity to N- and T-type Ca$_V$ channels[60–62]. In contrast, benidipine, in addition to L-type Ca$_V$ channel, can also inhibit both N-type and T-type Ca$_V$ channels, with IC$_{50}$ values of approximately 35 μM and 11 μM, respectively[59,63,64]. To understand the molecular basis of how benidipine is recognized by Ca$_V$3.1, we compared the benidipine-bound Ca$_V$1.2 structure with that of Ca$_V$3.1, and the result illustrates that most of the residues participating in benidipine binding are also conserved in Ca$_V$3.1, such as V1053$^{S5III}$, F1129$^{DIIIP1}$, and F1513$^{S6IV}$. Although I1046$^{S5III}$, M1178$^{S6III}$, F1181$^{S6III}$, and M1509$^{S6IV}$ in Ca$_V$1.2 are substituted by L1386$^{S5III}$, V1505$^{S6III}$, M1508$^{S6III}$, and L1813$^{S6IV}$ in Ca$_V$3.1, respectively, we speculate that these substitutions by similar residues are tolerable and that the relatively conserved binding pocket underlies the basis for how benidipine acts as an inhibitor in multiple voltage gated calcium channels (Fig. 4f). Moreover, two residues, T1056 and Q1060, play pivotal roles in the binding of DHP drugs to L-type Ca$_V$ channels. However, these residues are not conserved in N-type and T-type Ca$_V$. Residue T1056 of Ca$_V$1.2 is substituted by Y1289 in Ca$_V$2.2, and residue Q1060 is replaced by M1293 in Ca$_V$2.2 and F1400 in Ca$_V$3.1 (Supplementary Fig. 7a, b, Fig. 4f). Although benidipine does not inhibit mutations T1056Y, Q1060M, and Q1060F as potently as it does in WT Ca$_V$1.2, it still retains the ability to block the current

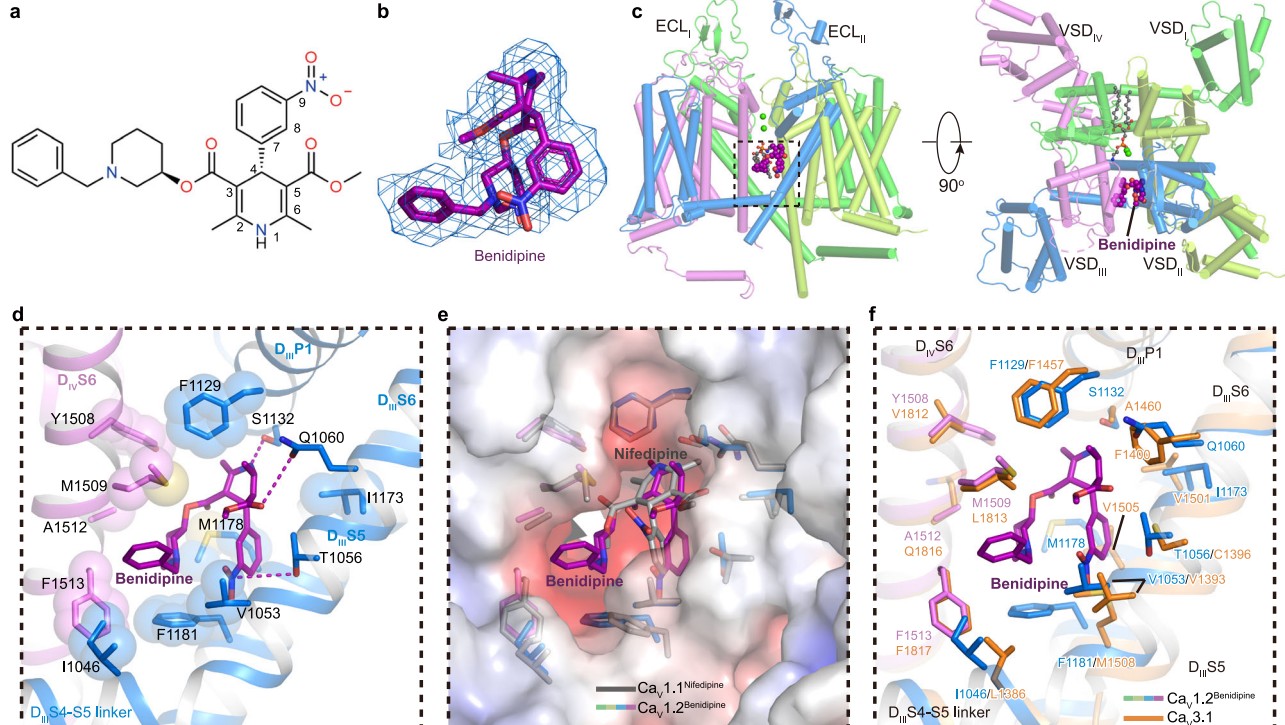

**Fig. 4 | Structure basis for inhibition of Ca$_V$1.2 by benidipine. a** Chemical structure of benidipine. **b** The cryo-EM density shown in blue mesh for benidipine in sticks. **c** The overall structure of the Ca$_V$1.2$^{BEN}$ complex. The domains of Ca$_V$1.2$^{BEN}$ are colored as D$_I$ in deep green, D$_{II}$ in light green, D$_{III}$ in deep blue, and D$_{IV}$ in mauve. The tetrandrine is presented as violet spheres. Two cation ions are shown as green spheres. Phospholipid entering through fenestration is shown as gray sticks. **d** Detailed binding sites for benidipine showing interactions between benidipine

and Ca$_V$1.2. The sidechains of key residues are displayed in sticks and the hydrophobic side chains are overlaid with transparent surfaces. Black dashed lines indicate potential hydrogen bonds. **e** Comparison of the DHP ligands binding sites of Ca$_V$1.2 with nifedipine bound Ca$_V$1.1 structure (PDB ID: 6JP5) (colored in gray), overlaid with the electrostatic surface potential of Ca$_V$1.2. **f** Comparison of the benidipine binding sites of Ca$_V$1.2 with Ca$_V$3.1 (PDB ID: 6KZO) (colored in orange).

(Supplementary Fig. 7c). In contrast, previous studies demonstrated that these same substitutions at positions T1056 and Q1060 almost abolished the sensitivity of Ca$_V$1.2 to the DHP drug R202-791 (ref. 65). The varied effects of these mutations on benidipine and R202-791 might arise from the phenylmethyl piperidinyl group in benidipine. This group establishes more interactions with the channel and appears to tolerate substitutions at positions T1056 and Q1060, especially in N- and T-type Ca$_V$ channels.

Several dihydropyridine drugs binding L-type calcium channels complex structures have been resolved in previous study[21,22,66–68], including amlodipine bound Ca$_V$Ab (7KMD), nimodipine bound Ca$_V$Ab (7KMF), nifedipine bound Ca$_V$1.1 (6JP5), amlodipine bound Ca$_V$1.1 (7JPX), (S)-Bay K 8644 bound Ca$_V$1.1 (6JP8) and (R)-Bay K 8644 bound Ca$_V$1.1 (7JPW). When Ca$_V$1.2$^{BEN}$ and these structures are superimposed, a same binding pocket in DIII-DIV fenestration are revealed to accommodate benidipine in Ca$_V$1.2 and amlodipine/(S)-Bay K 8644/ (R)-Bay K 8644 in Ca$_V$1.1 (Supplementary Fig. 8a–c). However, benidipine binding site do not overlap with binding sites of amlodipine and nimodipine in Ca$_V$Ab (Supplementary Fig. 8d–f), and the residues involved in interaction are not conserved as well.

L-type selective dihydropyridine (DHP) inhibitors and L/T-selective DHP inhibitors are cataloged in Supplementary Table 2. The structural similarities among Class 1, 2, and 3 necessitate collective consideration. Upon scrutinizing their chemical structures and inhibitory selectivity, we identified primary distinctions in functional group modifications, specifically additional alterations on the C3 and C4 benzene rings. Within Class 1 and Class 2, with the exception of three instances (cilnidipine, amlodipine, nimodipine), all DHP derivatives predominantly featured smaller, relatively hydrophilic C3 groups for L-type selective compounds, while L/T-type selective compounds

displayed larger and hydrophobic C3 groups. For Class 4 and Class 5 compounds, characterized by a modification into a ring at the C5/C6 position, the sole difference between the L-type selective Class 4-1 and L/T-type selective Class 4-4 compounds lies in the presence of a benzoic acid group on the C4 benzene ring in Class 4-1, while Class 4-4 has a hydroxyl group at the corresponding position. Thus, for compounds with C5-C6 rings, the hydrophobic nature of C3 modification does not exert a significant impact; the pivotal factor influencing differences is the presence of an additional aromatic group modification on the C4 benzene ring.

To further investigate interactions among compounds of varied selectivity on L-type and T-type channels, CaV1.2$^{BEN}$ served as a docking template. L-type selective compounds from Class 1 were docked onto L-type channel Ca$_V$1.2, while L/T-type selective compounds from Class 1 were docked onto both L-type channel Ca$_V$1.2 and T-type channel Ca$_V$3.1. Irrespective of L-type or L/T-type selectivity, these compounds bound to the D$_{III}$-D$_{IV}$ fenestration in Ca$_V$1.2 or Ca$_V$3.1, exhibiting a favorable superposition of the dihydropyridine backbone (Supplementary Fig. 9b, c). Both L-type selective and L/T-type selective DHPs docked to Ca$_V$1.2 formed hydrogen bonds with T1056, Q1060, and S1132. However, the C3 group of L-type selective DHPs docked to Ca$_V$1.2 is confined near M1509 within the hydrophobic pocket formed by D$_{III}$S6 and D$_{IV}$S6. In contrast, L/T-type selective DHPs docked to Ca$_V$3.1, while unable to form hydrogen bonds at the corresponding positions, not only occupy the binding pocket formed by D$_{III}$S6 and D$_{IV}$S6 but also extend to the hydrophobic pocket formed by D$_{III}$S5 and D$_{IV}$S5, akin to the position occupied by L/T-type selective DHPs docked to Ca$_V$1.2. This suggests that the ability of the C3 group to occupy the binding pocket formed by D$_{III}$S5 and D$_{IV}$S5 may be pivotal in determining the L-type and L/T-type selectivity of DHPs. In the case of

cilnidipine, its C3 benzenyl group occupies the C5 position in the Ca$_V$1.2 binding pocket, possibly due to the planar rigidity of the phenyl group preventing it from fitting into the C3 position. The excessively large C5 group may clash with F1400 in Ca$_V$3.1, elucidating why cilnidipine, despite its large and hydrophobic C3 group, cannot inhibit T-type channels. In the case of the other exception, amlodipine docked in Ca$_V$3.1, the C2 amino group extends towards the pore and forms potential interactions with hydrophilic residues K1462 and Q1868 near the pore, compensating for the modest interaction between the C3 group and Ca$_V$3.1. Given that L/T-type selective DHPs also exhibit different affinities for various T-type channels, such as nimodipine's preference for inhibiting Ca$_V$3.2 and Ca$_V$3.3 over Ca$_V$3.1 (ref. [64]), we aligned the Ca$_V$3.3 structure with the Ca$_V$3.1 structure. The alignment results revealed that the hydrophobic pockets formed by D$_{III}$S5 and D$_{IV}$S5 in Ca$_V$3.3 are smaller than those formed in Ca$_V$3.1, potentially explaining why nimodipine produces distinct inhibitory effects on T-type channels with different α1 subunits.

## Methods

### Electrophysiology

Whole-cell voltage-clamp recordings were carried out using human embryonic kidney 293 T (HEK293T) cells, which were cultured in Dulbecco's Modified Eagle Medium (DMEM, Gibco) supplemented with 10% (v/v) fetal bovine serum (FBS, PAN-Biotech) at 37 °C supplemented with 5% CO$_2$. Three distinct recombinant baculoviruses encoding Ca$_V$1.2, α2δ1, and β2 were used to coinfect HEK293T cells once they reached a confluence of 40–60%. After 8–12 h, transfected cells were re-plated on 8 × 8 mm coverslips, and sodium butyrate was added to the culture at a final concentration of 0.5–2.5 mM to improve protein expression. Whole-cell recordings of activation curves were performed from isolated GFP-positive HEK293T cells at 20–28 h after transfection with recombinant baculoviruses. Briefly, the external solution was composed of 105 mM NaCl, 30 mM TEA-Cl, 10 mM BaCl$_2$, 10 mM HEPES pH 7.3, 10 mM D-glucose, 1 mM MgCl$_2$, 5 mM CsCl, and 318 mOsm/L. The pipette solution contained 135 mM K-gluconate, 10 mM HEPES pH 7.2, 5 mM EGTA, 2 mM MgCl$_2$, 5 mM NaCl, 4 mM Mg-ATP, and 295 mOsm/L[69]. Pipettes were pulled to obtain final resistances of 2–6 MΩ with a Sutter P-97 puller and heat-polished before employment. For measuring the inhibitory effect of tetrandrine and benidipine, the series resistance was 2–10 MΩ and was compensated 80–90%; and for measuring the steady-state activation curve, the series resistance was <5 MΩ. The series resistance was not compensated in these experiments. All recordings were collected with an Axoclamp 700B amplifier and Digidata 1440 A (Molecular Devices). Signals were digitized at 10 kHz and low-pass filtered at 2 kHz. Data were analyzed with Clampfit 10.7.

To measure the inhibition curves, 2 µg endotoxin-free plasmids of wild-type (WT) or mutants generated by PCR expressing Ca$_V$1.2, α2δ1, and β2 were transiently transfected using 1.2 µg Lipofectamine 2000 Reagent (Thermo Fisher Scientific). For curves determining, re-plated cells were held at −100 mV, and then depolarized at +10 mV to elicit inward currents. To ensure stable currents during measurement, a series of traces were pre-recorded at a specific time interval without drug application for each construct. The duration of stable currents for WT, T1056Y, Q1060M, and Q1060F was determined as ~10 min, meanwhile the duration of stable currents for N741W and N1179W was determined as ~3 min. Consequently, the depolarization time interval is set every 1 min for benidipine and every 20 s for tetrandrine. Perfusions applied using a gravity-driven system. The inhibition curve of benidipine was determined by first perfusing with the external solution and then with various concentrations of benidipine until the current amplitude reached a steady-state level. The inhibition curve of tetrandrine was determined by applying serial dilutions of tetrandrine at different concentrations for 20 s each. Data analyses were performed using Origin 2022 (Origin Lab

Corporation), GraphPad Prism 9 (GraphPad Software, Inc.) and Adobe illustrator 2018. Inhibition curves were generated using a Hill equation.

$$\frac{I}{I_{max}} = \frac{1}{1 + (\frac{C}{IC_{50}})^H} \tag{1}$$

where $I$ is the current at different drug concentrations, $I_{max}$ is the maximal current of Ca$_V$1.2 without drug applied, $[C]$ is the concentration of drugs, IC$_{50}$ is the half-maximal inhibitory concentration and $H$ is the Hill coefficient.

### Clone, expression and purification of human Ca$_V$1.2-α2δ1-β2b complex

DNA fragments encoding Ca$_V$1.2 α1C, α2δ1, and β2b were amplified from a human cDNA library and subcloned into the pEG BacMam vector for co-expression in mammalian cells. For Ca$_V$1.2, the C-terminus was tandemly fused with a superfolder GFP (sfGFP) and a Twin-Strep affinity tag. The fluorescent protein mCherry and Twin-Strep tag were fused to the N-terminus of wild type α2δ1 and β2b subunits, respectively. Primers used for cloning and introducing mutations are provided in Supplementary Table 3. The Bac-to-Bac system (Invitrogen, USA) was used to produce recombinant baculovirus in sf9 cells (Thermofisher, 10902096). The HEK293F cells (Thermofisher, 11625019) at density of ~2 × 10$^6$ cells per ml were infected with 2% (v/v) P2 recombinant baculovirus and subsequently cultured at 37 °C with 5% CO$_2$. After 12 h, 10 mM sodium butyrate was added to the medium to enhance protein expression. The cells were harvested 60 h after infection.

The cell pellets were resuspended and grinded on ice in a Dounce homogenizer using buffer D containing 20 mM HEPES pH 7.5, 150 mM NaCl, 1 mM CaCl$_2$, 5 mM β-mercaptoethanol (β-ME), 2 µg/mL aprotinin, 1.4 µg/mL leupeptin, and 0.5 µg/mL pepstatin A. The membrane was enriched by centrifugation at 110,000 g for 40 min at 4 °C. The collected membrane pellets were solubilized in buffer D supplemented with 1% (w/v) n-dodecyl-β-D-maltoside (DDM), 0.15% (w/v) cholesteryl hemisuccinate (CHS) (Anatrace, USA), 2 mM adenosine triphosphate (ATP), and 5 mM MgCl$_2$ for 2 h at 4 °C. The solubilized membrane was subjected to centrifugation at 110,000 g for 40 min at 4 °C, and the supernatant was passed through a 0.22 µm filter before being applied to a streptavidin agarose column for protein affinity purification. Following loading of the sample, the column was washed with buffer D containing additional 0.025% DDM to remove nonspecific bound proteins. The Ca$_V$1.2 -α2δ1-β2b complex was then eluted with a buffer containing 5 mM desthiobiotin, 20 mM HEPES pH 7.5, 150 mM NaCl, 1 mM CaCl$_2$, 5 mM β-ME, and 0.007% glyco-diosgenin (GDN) (Anatrace, USA). The elution was concentrated to a final volume of 1 mL with 100-kDa Millipore Tube (Merck Millipore, Germany) and further purified through size exclusion chromatography (SEC) using a Superose 6 Increase 10/300 GL column (GE Healthcare, USA) equilibrated with a buffer containing 20 mM HEPES pH 7.5, 150 mM NaCl, 1 mM CaCl$_2$, 5 mM β-ME, and 0.007% GDN. The peak fractions at ~12.5 mL were collected and concentrated to 7 mg/mL for cryo-EM grids preparation. For the tetrandrine/benidipine bound Ca$_V$1.2 complex, 10 µM concentration of tetrandrine (MedChemExpress (Monmouth Junction, NJ, USA)) was supplied throughout the expression and purification of Ca$_V$1.2 complex and 1 µM concentration of benidipine (MedChemExpress (Monmouth Junction, NJ, USA)) was supplied in the membrane solubilization process. A final concentration of 100 µM tetrandrine/benidipine was added to the cryo-EM sample and incubated for 40 min on ice before application in grids.

### Cryo-EM sample preparation and data collection

Quantifoil 1.2/1.3 Cu 300 mesh grids were glow-discharged for 60 s under H$_2$-O$_2$ condition with a Solarus plasma cleaner (Gatan, USA)

before use. The grids were applied with a 2.5-μL droplet of protein sample at 4 °C and 100% humidity, and then snap-frozen in liquid ethane cooled by liquid nitrogen using a Vitrobot Mark IV (Thermo Fisher Scientific, USA). The grids sample was stored in liquid nitrogen before being checked on the electron microscopy instrument.

Cryo-EM data were collected on a 300-kV Titan Krios (Thermo Fisher Scientific, UAS) equipped with a K2 Summit direct electron detector (Gatan, USA) and a GIF-Quantum LS energy filter. The energy filter slit width was set to 20 eV. Movie stacks were acquired using SerialEM[70] at a calibrated magnification of 130,000× in the super-resolution mode, with defocus values ranging from −1.2 to −2.2 μm. The pixel size on motion-corrected micrographs was 1.04 Å. Each movie stack was dose-fractioned in 32 frames, yielding a total accumulated dose of 60 $e^-/Å^2$. The dose rate was set to 9.6 $e^-/pixel/s$.

### Cryo-EM data processing

For the data processing of $Ca_V1.2^{apo}$, a total of 1278 movie stacks were motion-corrected and dose-weighted using MotionCorr2 with 5 × 5 patches[71]. Contrast transfer function (CTF) estimation was performed using Gctf[72]. Particles that picked using Blob picker, Template picker, and Topaz picker in cryoSPARC[73] were combined, and duplicates were removed, yielding a total of 524k particles. Then, the particles were extracted into Relion 3.1 for guided multi-reference 3D classification against one good map which was generated by low pass filtering the high-resolution map of $Ca_V2.2$ (EMDB-31958) to 8 Å and 4 biased maps[74]. Particles from the class 1, which accounts for 47.1% of total particles, were selected and subjected to a single reference 3D classification to further remove poor particles, giving rise to one class with discernible structural features of α1, α2δ1, and β subunits as well as CTD. Particles belonging to this class were submitted for following 3D auto refinement, Bayesian Polish, and CTF refinement, yielding a 3.9-Å resolution map with clearly resolved transmembrane helices. To improve the quality of the map, a protein-only mask was used to avoid over-fitting of the detergent micelles in the subsequent 3D classification without particle alignment. The best class containing 35k particles was then imported back into cryoSPARC and subjected to Non-uniform Refinement, generating a final map reported at 3.5-Å resolution according to golden standard Fourier shell correlation (GSFSC) criterion.

A similar strategy was applied in the data processing of $Ca_V1.2^{TET}$ and $Ca_V1.2^{BEN}$. Specifically, a total of 449k and 337k particles were picked from 979 and 444 micrographs, respectively. The final maps of $Ca_V1.2^{TET}$ and $Ca_V1.2^{BEN}$ were reported at 3.4 Å and 3.3 Å, respectively. A diagram of data processing is summarized in Supplementary Fig. 2.

### Model building

To build the atomic model of $Ca_V1.2^{apo}$, we extracted rabbit α1, α2δ1, and human β3 subunits from the structures of the rabbit $Ca_V1.1$ complex (5GJV)[33] and $Ca_V2.2$ complex (7MIJ)[75] and generated homology models of the α1, α2δ1, and β2b subunits of $Ca_V1.2$ using phenix.sculptor program based on sequence alignment. The resulting models of three subunits were fitted into the map of the $Ca_V1.2^{apo}$ complex as rigid bodies using the UCSF Chimera. Then, the model was manually adjusted in COOT[76] iteratively, including the refinement of the main chain and side chains of residues. Phospholipid molecules were manually placed in the strip-shaped densities in both leaflets of the lipid bilayers and fenestrations of the pore domain of the $Ca_V1.2$ channel. Structure refinement was performed using phenix.real_space_refine application in PHENIX[77] in real space with secondary structure and Ramachandran restraints.

For the model building of $Ca_V1.2^{TET}$ and $Ca_V1.2^{BEN}$, the $Ca_V1.2^{apo}$ structure was used as the starting model and was fitted into the EM maps as a rigid body. The two-dimensional (2D) structures of tetrandrine and benidipine were downloaded from PubChem in SDF format, followed by the generation of 3D models and refinement restraints in

phenix.ligand_eLBOW. The drug molecules were docked into the EM map and refined according to the corresponding density. All the manually adjusted models were then subjected to real-space refinement using PHENIX.real_space_refine.

All figures were prepared with software PyMOLI[78] or UCSF Chimera[79].

### Docking

To investigate the interaction modes of various dihydropyridines (DHPs) in T-type and L-type calcium channels, a docking study was conducted utilizing the AutoDock Tools package (version 1.5.6)[80] and AutoDock Vina (version 1.1.2)[81]. In the docking simulation, L-type selective inhibitors (Nifedipine, Nitrendipine, Felodipine, Cilnidipine) and non-selective inhibitors (Benidipine, Nimodipine, Manidipine, Amlodipine)[64,82] were employed as ligands, while $Ca_V1.2$, $Ca_V3.1$ (6KZP)[83], and $Ca_V3.3$ (7WLI)[52] were utilized as receptors. The receptors and ligand were independently optimized and prepared as pdbqt format files required for docking. Given that the DHP binding pocket in the apo-state structure comprises only lipid and the site is considerably narrower compared to $Ca_V1.2$, the lipid and ligand was removed and the side chain dihedral angles of DHP binding pocket were adjusted. The docking grids for $Ca_V1.2$ were generated using enclosing boxes centered on benidipine, whereas the docking grids for $Ca_V3.1$ (or $Ca_V3.3$) were created using enclosing boxes centered on the residue L872 (or L769) involved in the DHP binding pocket. The processed ligands were subsequently docked into the two receptors. Only docking results with ligand conformations in a 'claw' shape configuration and biased towards DIII which analogous to the $Ca_V1.2$-benidipine complex that we have resolved were retained.

### Reporting summary

Further information on research design is available in the Nature Portfolio Reporting Summary linked to this article.

## Data availability

The data that support this study are available from the corresponding authors upon request. The three-dimensional cryo-EM density maps of the $Ca_V1.2$ complex in the apo state, tetrandrine bound state, and benidipine bound state have been deposited in the Electron Microscopy Data Bank under the accession codes EMD-34880, EMD-34891, and EMD-34892, respectively. The coordinates for the corresponding complexes have been deposited in Protein Data Bank under accession codes 8HLP ($Ca_V1.2^{apo}$), 8HMA ($Ca_V1.2^{TET}$), and 8HMB ($Ca_V1.2^{BEN}$). The model used for starting $Ca_V1.2$ complex model-building and docking is available in the PDB under the PDB ID 5GJV (rabit $Ca_V1.1$), 7MIJ (human $Ca_V2.2$), 6KZP (human $Ca_V3.1$ bound Z944), and 7WLI (human $Ca_V3.3$). Sequence of human $Ca_V1.2$ α1C, α2δ1 and β2b are available in Universal Protein Resource (Uniprot) databases under accession codes Q13936-1 [https://www.uniprot.org/uniprotkb/Q13936/entry] (CACNA1C), P54289-1 [https://www.uniprot.org/uniprotkb/P54289/entry] (CACNA2D1), and Q08289-3 [https://www.uniprot.org/uniprotkb/Q08289/entry] (CACNB2). The source data underlying Fig. 3f, Supplementary Figs. 1a, c, and 7c are provided as a Source Data file. Source data are provided with this paper.

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

## Acknowledgements

We thank Prof. X. Liu and Dr. J. Geng (Beihang University) for their invaluable assistance in the electrophysiological experiments. We thank X. Huang, B. Zhu, X. Li, and other staff members at the Center for Biological Imaging (CBI), Core Facilities for Protein Science at the Institute of Biophysics, Chinese Academy of Science (IBP, CAS) for the support in cryo-EM data collection. We thank Mr. Yan Wu for his research assistant service. This work was supported by the National Key Research and Development Program of China (grant 2021YFA1301501 to Y.Z.), the Chinese Academy of Sciences Strategic Priority Research Program (grant XDB37030304 to Y.Z., and grant XDB37030301 to X.C.Z.), the National Natural Science Foundation of China (grant 31971134 to X.C.Z., grant 92157102 to Y.Z., grant 82341246 and 82271498 to Z.H., and grant 32301026 to Y.D.), the Chinese National Programs for Brain Science and Brain-like Intelligence Technology (grant 2022ZD0205800 to Y.Z.),

Science and Technology Innovation 2030 Major Project (grant 2021ZD0202103 to Z.H.), Ningxia Hui Autonomous Region Key Research and Development Project (grant 2022BEG02042 to Z.H.), Beijing Chao-Yang Hospital Multi-disciplinary Team Program (grant CYDXK202211 to N.L.), and China Postdoctoral Science Foundation.

## Author contributions

Y.Z. conceived and supervised the project. Y.Q.W. carried out molecular cloning and cell biology experiments. Y.Q.W., X.W., and Y.D. expressed and purified protein complex samples for cryo-EM study. Y.Q.W., Z.Y., and Y.D. carried out cryo-EM data collection. Z.Y. and Y.Q.W. processed the cryo-EM data. Y.Q.W., Z.Y., and Y.H.W. built and refined the atomic model. Y.Z., Y.Q.W., and Z.Y. analyzed the structures. Y.Z. designed mutants and L.W., X.L. performed electrophysiological experiments. Q.B. designed and carried out molecule docking analysis. Y.Z., Z.Y., Y.Q.W., X.L., Q.B., R.L., Y.M., and H.X. prepared figures. N.L. summarized and revised diseases associated mutations in Supplementary data. Y.Q.W., Z.Y., and X.L. prepared the initial draft of the manuscript. Y.Z., X.C.Z., and Z.H. edited the manuscript with input from all authors in the final version.

## Competing interests

The authors declare no competing interests.
