## [Peer Review File · Nature Communications]

Structural Bases of Inhibitory Mechanism of CaV1.2 Channel InhibitorsReviewer #1 (Remarks to the Author):

The manuscript by Wei et al. presents the first structures of human CaV1.2 channel in the apo state, in the presence of the antirheumatic drug tetrandrine and in the presence of the antihypertensive drug benidipine. CaV1.2 plays an important role in muscle contractility, neuroplasticity, and neuroendocrine regulation, and is implicated in numerous cardiovascular and neurological disorders. Correspondingly, this channel represents an important drug target, making structure and function studies of CaV1.2 a promising avenue to help devising new therapeutic strategies. Given the high quality of structural work, the importance of CaV1.2 for human physiology and pathophysiology and novelty of presenting the first structures of the human ortholog in the absence or presence of clinically useful drugs, publication of this manuscript is undoubtedly well justified. The comments below are intended to further improve this important and impactful manuscript.

The study is clearly structure-centric but also includes nice examples of whole-cell patch-clamp recordings (Fig. 3, Supplementary Fig. 1). The functional technique should certainly be used more extensively to give a deeper insight into the structure and function of CaV1.2. In particular, it is important to provide the concentration dependencies of CaV1.2 inhibition by tetrandrine and benidipine. To validate physiological relevance of the identified binding sites of tetrandrine and benidipine, it is also important to make two-three mutations of key residues contributing to drug binding (e.g., N741, N1179 and Y1508 for tetrandrine and T1056, Q1060 and S1132 for benidipine), measure concentration dependencies of inhibition for these mutants and compare them to the concentration dependencies for wild-type channels. In this sense, the already presented data for a mutation of V400, which does not seem to be the most important residue contributing to tetrandrine binding, does not appear fully convincing.

When the apo state structure of CaV1.2 is described, it is referred to as representing an inactivated state (lines 108-109). Later, however, the S4 helices are referred to as representing the "activated" state (lines 116-117). These apparently contradicting statements need to be resolved by adding a discussion of gating conformations of the presented structures.

Minor comments

1. Lines 89-92. The text here makes an impression that the models of CaV1.2 describe the entire ion channel. However, later, when mapping disease mutations, the authors state that more than half of the mutations cannot be mapped because the corresponding regions are not resolved in the structures. Please provide the precise description of what regions have been included into structural models and which ones were not, specifying boundaries with residue numbers.
2. Line 99. "helices" should be "regions".
3. Lines 100-101. "two re-entrant short helices (P1 and P2 which connect S5 and S6) from each domain" should be "the re-entrant loops from each subunit that connect S5 and S6 and include P1 and P2 helices".
4. Lines 103-104. Reference to Supplementary Fig. 2-3 should probably be removed from here.
5. Line 117. "discrepancies" should probably be "differences".
6. Lines 118-121. It appears here that Supplementary Fig. 4b intends to illustrate shrinkage of the pore in CaV1.2 compared to CaV1.1. If so, this is not obvious. One way this can be done is by adding two panels (c and d) where the CaV1.2 and CaV1.1 pores are shown individually with semitransparent molecular surfaces. When the authors describe turning of the F394 and F1175 sidechains by 90 degrees, do they mean rotation of a section of S6 that accompanies an alpha-to-pi transition, similar to the one originally described in another tetrameric calcium channel, TRPV6 (doi: 10.1038/nature25182)?
7. Line 132. "Invisible due to conformational heterogeneity". Do authors mean that the corresponding regions have not been resolved in their structures?
8. Line 183. A lipid molecule is mentioned to penetrate through the fenestration site between DI and DII. This is a correct and careful description. However, in the Figure 3 legend, this lipid is specified as a phospholipid. How well the entire lipid is resolved to be specified as a phospholipid? If only an acyl chain is resolved, as it appears in the figure, the authors should be more careful and refer to this model as a lipid, an acyl chain, or a putative phospholipid.
9. Lines 184-190. Supplementary Fig. 6a is referred here as the one that illustrates distinct binding

mode of tetrandrine compared to the other drugs. However, superposition of the structures is cumbersome and does not resolve the individual binding modes or makes it easy to compare them. Showing the drugs in individual panels with the channel in the same orientation might solve this problem.

10. Line 230. Please fix the typo in naming the amino acid residue.

11. Line 239. Do authors mean that they used three different viruses to transduce HEK 293T cells?

12. Figure 2. To visualize whether the disease mutations cluster together in the context of the assembled channel, it would be nice along with the panels showing individual subunits to also introduce a panel(s) illustrating the entire channel with all 39 sites present.

13. Please describe the color coding of labels in Supplementary Fig. 2a.

14. The first sentence of the Supplementary Table 2 footnote. Do authors mean that the table sections are highlighted with the colors of structural domains, which were used, for example, in Figures 1 and 2?

Reviewer #2 (Remarks to the Author):

The authors present a cryo-EM structure of Cav1.2 in the absence and the presence of two different blocking molecules. Similarities to Cav1.1 structures are described, and the structural determinants of the interaction with the inhibitors is explored. The work with tetrandrine is interesting and it is a good choice for exploration, Benidipine is a bit of an odd choice, but given its inhibitory effect on T-type channels this offers some unique opportunities for discussion and insights into blocking mechanisms (see my comments below). The authors also map some of the disease linked mutations onto the structure, but I find this to be a distraction from the focus of the paper. But great job overall!

Points to address:

There is already a bioarchives paper on Cav1.2 that has been published by Dan Minor's group in Oct 2022 – this could be discussed here or at least cited.

The mapping of the mutations is not particularly useful as the authors do not clearly relate this mechanistically to their functional effects (it is mostly speculation). It would be more useful if the authors had obtained a structure of the mutant channels, or else used molecular analyses to predict how the mutations alter the 3D configuration of the channel. Perhaps Fig 2 can be moved into the supplemental section or deleted altogether as it is not pertinent to the pharmacology that is studied here (and saved for another paper).

The tetrandrine work is quite nice, but I find that the discussion of the data is not adequate – there is no discussion of the seminal paper by the Catterall lab on CavAB inhibition by DHPs (Tang et al Nature 2016), nor is there a detailed enough discussion of the senior author's prior work on Cav1.1 modulation by DHPs (Zhao et al, Cell) – the paper is cited, but it is discussed only in passing. Moreover, there is an opportunity for a much better discussion on DHPs that block only L-type channels versus those that act on both T-types and L-types. For example, nimodipine is a good blocker of T-types, but nifedipine is not. Also consider PMID: 24149495, PMID: 24990197, PMID: 30408648, PMID: 36209630, PMID: 31419643, PMID: 11854446 as examples of DHP compounds that block both L-type and T-type channels – can the authors come up with an explanation as to why these molecules block Cav1.2 and Cav3 channels based on their chemical structures in comparison with benidipine and how it interacts with the two channel subtypes? This could be quite easily done with an in silico docking program, or even by a crude comparison of these various chemical structures. In my view, this would be much more useful than what is shown in Figure 2 and a much better focus with the potential for much greater impact.

Grammar could be improved – there are numerous errors

Supplemental Fig 1a is not needed – we all know what a Cav1.2 channel looks like

electrophysiologically

Supplemental Fig 5 may not be needed if Figure 2 is eliminated

Point-by-point response for

Structural Bases of Inhibitory Mechanism of Cav1.2 Channel Inhibitors

TO REVIEWER #1

*The manuscript by Wei et al. presents the first structures of human Cav1.2 channel in the apo state, in the presence*
*of the antirheumatic drug tetrandrine and in the presence of the antihypertensive drug benidipine. Cav1.2 plays an*
*important role in muscle contractability, neuroplasticity, and neuroendocrine regulation, and is implicated in*
*numerous cardiovascular and neurological disorders. Correspondingly, this channel represents an important drug*
*target, making structure and function studies of Cav1.2 a promising avenue to help devising new therapeutic*
*strategies. Given the high quality of structural work, the importance of Cav1.2 for human physiology and*
*pathophysiology and novelty of presenting the first structures of the human ortholog in the absence or presence of*
*clinically useful drugs, publication of this manuscript is undoubtedly well justified. The comments below are*
*intended to further improve this important and impactful manuscript.*

**Reply:** We greatly appreciate the reviewer's positive comments.

*The study is clearly structure-centric but also includes nice examples of whole-cell patch-clamp recordings (Fig. 3,*
*Supplementary Fig. 1). The functional technique should certainly be used more extensively to give a deeper insight*
*into the structure and function of Cav1.2. In particular, it is important to provide the concentration dependencies of*
*Cav1.2 inhibition by tetrandrine and benidipine.*

**Reply:** We totally agree with the reviewer's suggestions to incorporate a more systematic functional
study. In the revised manuscript, we included the concentration-dependent curves for tetrandrine and
benidipine applied to the wild-type Cav1.2 and its mutants in Figure 3f and Supplementary Figure 7c,
respectively. Both compounds demonstrate a concentration-dependent inhibition of the Cav1.2 current,
with a notable 90% current reduction observed at 10 μM for tetrandrine and 100 μM for benidipine.

We also attached the related figures here for your review.

Fig. 3f

Supplementary Fig. 7c

**Figure 1*** Concentration-dependent curves for the inhibition of tetrandrine (a) and benidipine (b) on
both Cav1.2^{WT} and its mutants. (Corresponding to Fig. 3f and Supplementary Fig. 7c)

*To validate physiological relevance of the identified binding sites of tetrandrine and benidipine, it is also important*
*to make two-three mutations of key residues contributing to drug binding (e.g., N741, N1179 and Y1508 for*
*tetrandrine and T1056, Q1060 and S1132 for benidipine), measure concentration dependencies of inhibition for*
*these mutants and compare them to the concentration dependencies for wild-type channels. In this sense, the*

*already presented data for a mutation of V400, which does not seem to be the most important residue contributing*
*to tetrandrine binding, does not appear fully convincing.*

**Reply:** We appreciate the reviewer's comment and agree that mutating key residues involved in drug
binding would be insightful. In the revised manuscript, we introduced mutants N741W and N1179W,
which are anticipated to create steric hindrance between tetrandrine and Cav1.2. The mutants T1056Y,
Q1060M, and Q1060F were engineered based on the sequence alignment of Cav1.2, Cav2.2, and
Cav3.1. These mutations are crucial for understanding the selectivity of benidipine among distinct types
of Cav channels.

During the determination of the dose-response curve, we observed a significant run-down effect
(reduction of the current), which considerably delayed the preparation of this response. To eliminate the
influence of the run-down effects on the inhibitory response of drug, we optimized our measurement
protocol by identifying the time interval where the current remained stable. All measurements were then
conducted exclusively within this identified stable period. The original V400W is not in the revised
manuscript due to severe run-down effects. Here, we provide the current traces for both wild-type and
mutant Cav1.2 observed during the stable window.

**Figure 2*** Representative traces for Cav1.2^{WT}, Cav1.2^{N741W}, and Cav1.2^{N1179W}.

The duration of stable currents for N741W and N1179W was determined as ~3 min. The “20 s” label
indicates the pre-measurement holding time. Scale bar at the bottom right indicates the time and current scale
of the trace.

Figure 3* Representative traces for Cav1.2^{WT}, Cav1.2^{T1056Y}, Cav1.2^{Q1060F}, and Cav1.2^{Q1060M}.

The duration of stable currents for T1056Y, Q1060F, and Q1060M was determined as ~10 min. The “1 min” and “4 min” labels indicate the pre-measurement holding time. Scale bar at the bottom right indicates the time and current scale of the trace.

Using the optimized method, the dose-response curves for tetrandrine in both mutants exhibited a significant rightward shift compared to those of wild-type Cav_v1.2. The determined IC₅₀ values for tetrandrine on wild-type, mutants N741W and N1179W were 14.2 nM, 71.3 nM, and 41.8 nM, respectively. These IC₅₀ values for mutants were distinguished from that of the wild-type, suggesting that N741 and N1179 are important residues for tetrandrine binding. The inhibition was not abolished by these two mutants, possibly because a single point mutation might not be sufficient to disrupt the extensive hydrophobic contacts between tetrandrine and Cav_v1.2.

In addition to the L-type Cav channels, benidipine also demonstrates a pronounced inhibitory effect on N- and T-types of Cav channels, with IC₅₀ values of 35 μM and 11 μM, respectively (T. Furukawa et al., 1999). We speculate that the relatively reduced potencies of benidipine on N- and T-type Cav channels are attributed to sequence variations in the binding pocket. Specifically, T1056 is substituted by Y1289 in Cav_v2.2, while Q1060 is replaced by M1293 in Cav_v2.2 and by F1400 in Cav_v3.1. We engineered three mutants: T1056Y, Q1060M, and Q1060F. The dose-response curves of these mutants displayed a marked rightward shift relative to the wild-type Cav_v1.2, yielding IC₅₀ values of 7.5 μM for Q1060M, 11.7 μM for Q1060F, and exceeding 100 μM for the T1056Y mutant. These findings not only consistent with the observed sensitivity of N- and T-Cav to benidipine but also underscore the critical role of residues T1056 and Q1060 in benidipine binding.

**Figure 4*** Concentration-dependent curves for the inhibition of tetrandrine (a) and benidipine (b) on
 both Cav1.2^{WT} and its mutants.

The revised Figure 3 and Supplementary Figure 7 are attached below for your convenience.

**Figure 5*** Structure basis for blockade of Cav1.2 by tetrandrine. (Corresponding to Fig. 3)

**a.** Chemical structure of tetrandrine. **b.** The cryo-EM density shown in blue mesh for tetrandrine in sticks. **c.**
 The overall structure of the Cav1.2^{TET} complex. The domains of Cav1.2^{TET} are colored as D_I in deep green,
 D_{II} in light green, D_{III} in deep blue, and D_{IV} in mauve. The tetrandrine located in the pore domain is presented
 as yellow spheres. **d-e.** Detailed binding sites for tetrandrine showing interactions between tetrandrine and
 Cav1.2. The sidechains of key residues are displayed in sticks and the hydrophobic side chains are overlaid
 with transparent surfaces. Black dashed lines indicate potential hydrogen bonds. A lipid located within the
 fenestration site is shown in gray. **f.** Dose-response curves of tetrandrine inhibition on Cav1.2^{WT}, Cav1.2^{N741W},
 and Cav1.2^{N1179W}. Cav1.2^{N741W} and Cav1.2^{N1179W} shift the IC₅₀ of tetrandrine on Cav1.2 from 14.2 nM to 71.3
 nM and 41.8 nM, respectively. Data in curves are obtained from 10 biologically independent experiments and
 represented as mean ± SEM.

**Figure 6* Reduced inhibition of benidipine on Cav1.2 mutants. (Corresponding to Supplementary Fig. 7)**

**a.** Sequence alignment of benidipine binding sites on Cav1.2, Cav2.2, and Cav3.1. Residues contributed to
 forming hydrogen bonds with benidipine are indicated by red triangles. Conserved alkaline and uncharged
 residues are shaded in blue and grey, respectively. **b.** Comparison of the benidipine binding sites of Cav1.2
 with Cav2.2 (PDB ID: 7VFS, colored in gray). **c.** Dose-response curves of benidipine on Cav1.2, Cav1.2^{T1056Y},
 Cav1.2^{Q1060M}, and Cav1.2^{Q1060F}. The IC₅₀ of Cav1.2^{WT}, Cav1.2^{T1056Y}, Cav1.2^{Q1060M}, and Cav1.2^{Q1060F} are 216.9
 nM, 267.5 μM, 7.5 μM, and 11.7 μM, respectively. Data in curves are obtained from 4 biologically
 independent experiments and represented as mean ± SEM.

We also included discussions in the revised manuscript. In the line 207, it reads “To validate the binding
 site of tetrandrine, we designed two mutants, N741W and N1179W, with the potential to create steric
 clash between tetrandrine and the binding site. We determined the dose-response curves of tetrandrine
 for both wild-type and these two mutants. The curves for both mutants exhibited a rightward shift
 compared to that of wild-type Cav1.2, indicating that these mutants reduce the sensitivity of these
 mutants to tetrandrine. The derived IC₅₀ values for tetrandrine with the N741W and N1179W mutants are
 71.3 nM and 41.8 nM, respectively, which are higher than that of wild-type (~14.2 nM) (Fig. 3), further
 confirming the binding site of tetrandrine.”.

[revised manuscript text omitted]

*When the apo state structure of Cav1.2 is described, it is referred to as representing an inactivated state (lines 108-*
*109). Later, however, the S4 helices are referred to as representing the “activated” state (lines 116-117). These*
*apparently contradicting statements need to be resolved by adding a discussion of gating conformations of the*
*presented structures.*

**Reply:** Thanks for your kind reminder. To avoid confusion, we used the term "up" conformation to
describe the S4 segment in the revised manuscript, which is also a conventional terminology.
Furthermore, we have incorporated a discussion on the gating conformations of the presented structures,
as suggested by the reviewer, in lines 114. It states: "Although all S4 helices in the four VSDs are in an
'up' conformation, the intracellular gate remains closed, suggesting that the structure of Ca_v1.2 was
captured in an inactivated state."

**Minor comments:**

*1. Lines 89-92. The text here makes an impression that the models of CaV1.2 describe the entire ion channel.*
*However, later, when mapping disease mutations, the authors state that more than half of the mutations cannot be*
*mapped because the corresponding regions are not resolved in the structures. Please provide the precise*
*description of what regions have been included into structural models and which ones were not, specifying*
*boundaries with residue numbers.*

**Reply:** We thank the reviewer for this comment and have provided the exact range of structures we
analyze and build in the figure legend of Figure 1. It now reads: “The α_1 subunit is colored in deep green
(D_I: A114-R211, F231-I447), light green (D_{II}: W509-K778), deep blue (D_{III}: F889-I943, F964-Y1197),
mauve (D_{IV}: K1198-S1321, E1350-T1357, S1372-E1484, P1494-D1533), and orange (CTD: W1534-
P1574, C1582-S1593, N1607-K1638, K1647-P1655), respectively. The β_2b (D222-R276, L297-W407),
α_2 (F27-D129, Q143-V224, K232-N540, Q554-K690, N698-T828, D849-P906), and δ_1 (Q972-G1022,
T1028-D1070) subunits are colored in magenta, turquoise, and pink, respectively.”.

*Line 99. “helices” should be “regions”.*

**Reply:** We thank the reviewer for pointing out this. We have replaced “helices” with “regions” at the
original line 99.

It now reads: “(Line 103) The S5-S6 regions of all four domains form the pore domain to facilitate and
control the passage of ions (Fig. 1c–d).”.

*2. Lines 100-101. “two re-entrant short helices (P1 and P2 which connect S5 and S6) from each domain” should*
*be “the re-entrant loops from each subunit that connect S5 and S6 and include P1 and P2 helices”.*

**Reply:** We thank the reviewer for better expression. We have replaced corresponding phrase at the
original line 100-101.

It now reads: “(Line 105) The selectivity filter (SF) is contributed by the re-entrant loops from each
subunit that connect S5 and S6 and include P1 and P2 helices (Fig. 1e).”.

*3. Lines 103-104. Reference to Supplementary Fig. 2-3 should probably be removed from here.*

**Reply:** We thank the reviewer for pointing out this. We have removed the “Supplementary Fig. 2-3” from
reference in original lines 103-104.

It now reads “(Line 106) In contrast to the conformational heterogeneity of the α -interaction domain (AID)
found in Ca_v1.1 (Ref. ³³), AID is well resolved in the current Ca_v1.2 structure, with its N- and C-termini
adjacent to the intracellular gate and VSD_{II}, respectively (Fig. 1a).”.

*5. Line 117. “discrepancies” should probably be “differences”.*

**Reply:** We thank the reviewer for pointing out this. We have replaced “discrepancies” with “differences”
at the original line 117.

It now reads: “(Line 125) However, some structural differences are observed between the structures of
Ca_v1.2^{apo} and Ca_v1.1.”.

*6. Lines 118-121. It appears here that Supplementary Fig. 4b intends to illustrate shrinkage of the pore in Ca_v1.2*
*compared to Ca_v1.1. If so, this is not obvious. One way this can be done is by adding two panels (c and d) where*
*the Ca_v1.2 and Ca_v1.1 pores are shown individually with semitransparent molecular surfaces. When the authors*
*describe turning of the F394 and F1175 sidechains by 90 degrees, do they mean rotation of a section of S6 that*
*accompanies an alpha-to-pi transition, similar to the one originally described in another tetrameric calcium*
*channel, TRPV6 (doi: 10.1038/nature25182)?*

**Reply:** Thank you very much for this comment. In fact, our intention was not to illustrate a reduction in
pore size in Ca_v1.2 compared to Ca_v1.1. While there are local conformational changes in the S6 helices,
the intracellular gates of these channels remain almost identical. However, these local alterations result

in a reduced central cavity size. To avoid potential misunderstandings, we have incorporated additional
 panels in Supplementary Fig. 4, highlighting the closed intracellular gate and the radii of the full pores.
 Moreover, the discussion was modified in line 126, it now reads: “For example, two π -bulges occur on
 the S6^{DI} and S6^{DIII} helices of Cav1.2^{apo}, whereas both helices assume a canonical α -helix conformation in
 Cav1.1. The sidechains of residues on the intracellular segment, starting from F394^{S6I} and F1175^{S6III},
 undergo an approximate 90° rotation (Supplementary Fig. 4b). The residues forming the central cavity
 and the intracellular gate differs from that in Cav1.1. In specific, the side chain of F394 and F1175 face
 the central cavity, causing a shrinkage of the inner radius of the central cavity (Supplementary Fig. 4c.d).
 In Cav1.1, the intracellular gate is constituted by V329, L333, F656, A660, F1060, V1064, F1376, and
 I1380. With the local conformational shifts on the S6 helices in Cav1.2, the intracellular gate is formed by
 L401, S405, L749, V753, V1182, I1186, F1519, and I1523 (Supplementary Fig. 4e). A previous study on
 TRPV6 channel suggested that the transition from an α -helix to a π -helix may prompt channel opening
 (McGoldrick et al., 2018). However, despite such transition being observed in Cav1.2, its intracellular
 gate remains closed (Supplementary Fig. 4f).”

Figure 7* Structural comparison of the Cav1.2^{apo} with Cav1.1 structure. (Corresponding to Supplementary Fig. 4)

**a.** Structural comparison of the overall structure of Cav1.2^{apo} and Cav1.1 (PDB ID:5GJW). **b.** Secondary
structural transition in the middle of S6_I and S6_{III}. The helical turns consisting of residues L390-F394
and I1172-F1175 shift from α to π helix. **c.d.** Extracellular view of gate composed of S6 in Cav1.1 (c)
and Cav1.2 (d). **e.** The intracellular gate formed by four S6 helices viewed from the extracellular side. **d.**
Plot of pore radii for the Cav1.1(PDB ID:5GJW) and Cav1.2^{apo} complex.

*7. Line 132. "Invisible due to conformational heterogeneity". Do authors mean that the corresponding regions*
*have not been resolved in their structures?*

**Reply:** We appreciate the reviewer's comment. N-terminal, C-terminal and cytosolic linkers between
different domains have not been resolved in our or previous Ca_v structures.

To avoid confusion, we have modified this sentence. In the line 146, it now reads "The remaining 47
mutation sites are mainly distributed on the N-terminus, cytosolic long linkers between different domains
(mainly DII-DIII), and the C-terminal tail, and they were not determined due to conformational
heterogeneity."

Additionally, we also described the regions we determined in the legend of figure legend 1. The revised
legend reads "The overall EM density map and structure of the Cav1.2 complex. The α 2 δ 1 and β 2b
subunits, voltage-sensing domain (VSD), C-terminal domain (CTD), α -interacting domain (AID), and N-
glycans were labeled. The α 1 subunit is colored in deep green (D_I: A114-R211, F231-I447), light green
(D_{II}: W509-K778), deep blue (D_{III}: F889-I943, F964-Y1197), mauve (D_{IV}: K1198-S1321, E1350-T1357,
S1372-E1484, P1494-D1533), and orange (CTD: W1534-P1574, C1582-S1593, N1607-K1638, K1647-
P1655), respectively. The β 2b (D222-R276, L297-W407), α 2 (F27-D129, Q143-V224, K232-N540, Q554-
K690, N698-T828, D849-P906), and δ 1 (Q972-G1022, T1028-D1070) subunits are colored in magenta,
turquoise, and pink, respectively. N-glycans are colored in yellow."

*8. Line 183. A lipid molecule is mentioned to penetrate through the fenestration site between DI and DII. This is a*
*correct and careful description. However, in the Figure 3 legend, this lipid is specified as a phospholipid. How well*
*the entire lipid is resolved to be specified as a phospholipid? If only an acyl chain is resolved, as it appears in the*
*figure, the authors should be more careful and refer to this model as a lipid, an acyl chain, or a putative*
*phospholipid.*

**Reply:** We agree with the reviewer and have rephrased our statement according to this comment. In the
revised Figure 3 legend, it reads "A lipid located within the fenestration site is shown in gray."
Additionally, to clearly illustrate the presence of the lipid molecules, we have included density maps of
these lipid molecules located at the fenestration site between DI and DII in Cav1.2^{apo} (left), Cav1.2^{TET}
(center), and Cav1.2^{BEN} (right) in Supplementary Fig. 3b. The revised Supplementary Fig. 3 is attached
below for your convenience.

Figure 8* Cryo-EM map of the $Ca_v1.2$ structure. (Corresponding to Supplementary Fig. 3)

a. The cryo-EM density map and atomic model of S1-S6 segments in the four repeats of $Ca_v1.2^{apo}$. The side chains of key residues are labeled. The cryo-EM maps are shown as blue mesh. **b.** The cryo-EM density maps of the lipid molecules entering the fenestration site between DI and DII in $Ca_v1.2^{apo}$ (left), $Ca_v1.2^{TET}$ (center), and $Ca_v1.2^{BEN}$ (right).

9. Lines 184-190. Supplementary Fig. 6a is referred here as the one that illustrates distinct binding mode of
tetrandrine compared to the other drugs. However, superposition of the structures is cumbersome and does not
resolve the individual binding modes or makes it easy to compare them. Showing the drugs in individual panels
with the channel in the same orientation might solve this problem.

**Reply:** We appreciate this comment to enhance clarity in this figure. We have added four panels to show
the unique binding modes of tetrandrine in comparison to other drugs. For ease of comparison, all
panels maintain the same orientation. The updated Supplementary Fig. 6 is attached below for your
reference.

**Figure 9* Binding mode of pore blocker tetrandrine compared with other antagonists. (Corresponding to**
**Supplementary Fig. 6)**

**a.** The binding mode of tetrandrine (TET) compared with other antagonists. Cav1.2^{TET} is shown as scaffold
for comparison, tetrandrine is colored yellow. The Cav1.1 structure bound dihydropyridine class nifedipine
(DHP, colored in pink) (PDB ID: 6JP5), phenylalkylamine class verapamil (PAA, colored in blue) (PDB ID:
6JPA), benzothiazine class diltiazem (BTZ, colored in wheat) (PDB ID: 6JPB), and Cav1.3 structure bound
cinnarizine (CIN, colored in grey) (PDB ID: 7UHF) were aligned to Cav1.2^{TET}. **b-e.** The binding site of
tetrandrine compared with nifedipine, verapamil, cinnarizine, and diltiazem. **f.** Tetrandrine occupies the
binding site of diltiazem.

10. Line 230. Please fix the typo in naming the amino acid residue.

**Reply:** We appreciate reviewer for pointing out this typo. The “phenalene” should be replaced to be
“phenylalanine”. It has been corrected in revised manuscript.

11. Line 239. Do authors mean that they used three different viruses to transduce HEK 293T cells?

**Reply:** Thanks for the comments. In electrophysiology experiment, three different viruses were added
simultaneously to infect HEK 293T cells. To clarify this operation, we revised this sentence. In the line

318, it reads “Three distinct recombinant baculoviruses encoding Cav1.2, $\alpha\delta 1$, and $\beta 2$ were used to
coinfect HEK293T cells once they reached a confluence of 40-60%.”.

12. Figure 2. To visualize whether the disease mutations cluster together in the context of the assembled channel, it
would be nice along with the panels showing individual subunits to also introduce a panel(s) illustrating the entire
channel with all 39 sites present.

**Reply:** We would like to express our appreciation for this suggestion to modify Figure 2. We added a
panel (panel a) in Figure 2 to introduce the distribution of 39 mutations in the entire channel. The revised
Figure 2 is attached below.

**Figure 10* Molecular mapping of pathogenic mutations. (Corresponding to Fig. 2)**

a-f Disease related mutations are located on D_I (b), D_{II} (c), D_{III} (d), D_{IV} (e), and CTD (f). Mutation sites are
marked with spheres and colored in blue (gain of function), yellow (loss of function), and red (other),
respectively.

13. Please describe the color coding of labels in Supplementary Fig. 2a.

**Reply:** We thank the reviewer for pointing out this. We have added the color coding of labels in the figure
legend of Supplementary Fig. 2a. It now reads “a. Flowchart of cryo-EM data processing. Representative
raw cryo-EM micrograph and 2D class averages shown distinct secondary structure features from
different views of Cav1.2^{apo}, respectively (Bar = 400 Å). Several rounds of 3D classifications are
conducted to clean particles, followed by Bayesian Polish and CTF Refine to improve image quality. The
resolution and number of particles during data processing are labeled (Cav1.2^{apo} in black, Cav1.2^{TET} in
orange, and Cav1.2^{BEN} in purple) and details can be found in Materials and Methods. The final map was
reported at 3.5-Å for Cav1.2^{apo}, 3.4-Å for Cav1.2^{TET}, and 3.3-Å for Cav1.2^{BEN} according to the GSFSC
criterion, respectively.”. At the same time, we added some text on the figure for more clarity. The revised
Supplementary Figure 2 is attached below.

**Figure 11* Cryo-EM data processing of $\text{Cav}1.2^{\text{ap0}}$, $\text{Cav}1.2^{\text{TET}}$, and $\text{Cav}1.2^{\text{BEN}}$. (Corresponding to**
 **Supplementary Fig. 2)**

**a.** Flowchart of cryo-EM data processing. Representative raw cryo-EM micrograph and 2D class averages
 shown distinct secondary structure features from different views of $\text{Cav}1.2^{\text{ap0}}$, respectively (Bar = 400 Å).
 Several rounds of 3D classifications are conducted to clean particles, followed by Bayesian Polish and CTF

Refine to improve image quality. The resolution and number of particles during data processing are labeled
(Cav1.2^{apo} in black, Cav1.2^{TET} in orange, and Cav1.2^{BEN} in purple) and details can be found in Materials and
Methods. The final map was reported at 3.5-Å for Cav1.2^{apo}, 3.4-Å for Cav1.2^{TET}, and 3.3-Å for Cav1.2^{BEN}
according to the GSFSC criterion, respectively. **b.** Electron density map colored by local resolution values.
From left to right: Cav1.2^{apo}, Cav1.2^{TET} and Cav1.2^{BEN}. **c.** Fourier Shell Correlations (FSC) of the final map
of Cav1.2^{apo} (left), Cav1.2^{TET} (middle) and Cav1.2^{BEN} (right), calculated between two independently refined
half-maps before (blue) and after (red) post-processing. The FSC curve calculated between the cryo-EM
density map and the structural model are shown in black.

*14. The first sentence of the Supplementary Table 2 footnote. Do authors mean that the table sections are*
*highlighted with the colors of structural domains, which were used, for example, in Figures 1 and 2?*

**Reply:** We appreciate the reviewer's comment. The highlighted colors used in Supplementary Table 2
are in keeping with the colors of structural domains, but we adjusted the opacity of these colors for
aesthetics.

**TO REVIEWER #2**

*The authors present a cryo-EM structure of Cav1.2 in the absence and the presence do two different blocking*
*molecules. Similarities to Cav1.1 structures are described, and the structural determinants of the interaction with*
*the inhibitors is explored. The work with tetrandine is interesting and it is a good choice for exploration,*
*Benipidine is a bit of an odd choice, but given its inhibitory effect on T-type channels this offers some unique*
*opportunities for discussion and insights into blocking mechanisms (see my comments below). The authors also*
*map some of the disease linked mutations onto the structure, but I find this to be a distraction from the focus of the*
*paper. But great job overall!*

**Reply:** We appreciate the reviewer's positive comments very much.

**Points to address:**

*There is already a bioarchives paper on Cav1.2 that has been published by Dan Minor's group in Oct 2022 – this*
*could be discussed here or at least cited.*

**Reply:** Thank you very much for your comments. The citation has also been added in the revised
manuscript. In the line 119 it now reads: "A lipid molecule was observed in the D_I-D_{II} fenestration of
Cav1.2^{apo}, consistent with another structural study on Cav1.2 channel (Ref.^{34,35}).".

*The mapping of the mutations is not particularly useful as the authors do not clearly relate this mechanistically to*
*their functional effects (it is mostly speculation). It would be more useful if the authors had obtained a structure of*
*the mutant channels, or else used molecular analyses to predict how the mutations alter the 3D configuration of the*
*channel. Perhaps Fig 2 can be moved into the supplemental section or deleted altogether as it is not pertinent to*
*the pharmacology that is studied here (and saved for another paper).*

*Supplemental Fig 5 may not be needed if Figure 2 is eliminated.*

**Reply:** We appreciate the suggestion. In deed, much of the content related to Figure 2 is speculative,
and more comprehensive studies are essential to thoroughly understand the impact of the mutants.
Nevertheless, at least 86 mutations of Cav1.2 α 1 subunit have been identified in various cases of human
diseases, and in this paper, the structure of Cav1.2 was determined, and these mutation sites are
localized for the first time. We keep the discussion of Figure 2 and Supplemental Fig 5 to provide

relevant researchers with as much new structural information as possible.

*The tetrandine work is quite nice, but I find that the discussion of the data is nor adequate – there is no discussion*
*of the seminal paper by the Catterall lab on CavAB inhibition by DHPs (Tang et al Nature 2016), nor is there a*
*detailed enough discussion of the senior author’s prior work on Cav1.1 modulation by DHPs (Zhao et al,Cell) –*
*the paper is cited, but it is discussed only in passing.*

**Reply:** Thanks for your kind reminder, which will make the manuscript more comprehensive. In the
revised manuscript, we aligned benidipine bound Cav_v1.2 with amlodipine/nimodipine bound Cav_vAb and
amlodipine/(S)-Bay K 8644/(R)-Bay K 8644 bound Cav_v1.1 (Supplementary Fig. 8). The binding site of
benidipine in DIII-DIV fenestration overlapped with binding site of other DHPs in Cav_v1.1, but was not
coincide with binding site of amlodipine/nimodipine in Cav_vAb. The residues involved in forming hydrogen
bonds between benidipine and Cav_v1.2 are conserved in Cav_v1.1, but not conserved in Cav_vAb. Based on
these observation, a discussion was added in manuscript (Line 264): “Several dihydropyridine drugs
binding L-type calcium channels complex structures have been resolved in previous study²¹.(Tang et al.,
2014; Tang et al., 2016; Yao, Gao, & Yan, 2022; Zhao et al., 2019), including amlodipine bound Cav_vAb
(7KMD), nimodipine bound Cav_vAb (7KMF), nifedipine bound Cav_v1.1 (6JP5), amlodipine bound Cav_v1.1
(7JPX), (S)-Bay K 8644 bound Cav_v1.1 (6JP8) and (R)-Bay K 8644 bound Cav_v1.1 (7JPW). When
Cav_v1.2^{BEN} and these structures are superimposed, a same binding pocket in DIII-DIV fenestration are
revealed to accommodate benidipine in Cav_v1.2 and amlodipine/(S)-Bay K 8644/(R)-Bay K 8644 in Cav_v1.1
(Supplementary Fig. 8a-c). However, benidipine binding site do not overlap with binding sites of
amlodipine and nimodipine in Cav_vAb (Supplementary Fig. 8d-f), and the residues involved in interaction
are not conserved as well.”.

**Figure 12* Comparison of benidipine and DHPs resolved in Cav_vAb and Cav_v1.1. (Corresponding to**
**Supplementary Fig. 8)**

**a-c.** The binding site of benidipine compared with amlodipine / (S)-Bay K 8644 / (R)-Bay K 8644 in Cav_v1.1
(PDB ID: 7JPX / 6JP8 / 7JPW). **d.** The binding sites of benidipine compared with other DHPs resolved in

CavAb. Cav1.2^{BEN} is shown as scaffold for comparison, benidipine is colored in purple. The CavAb structure
bound nimodipine (PDB ID: 5KMF) and amlodipine (PDB ID: 5KMD) were aligned to Cav1.2^{BEN}. e-f. The
binding site of benidipine compared with nimodipine and amlodipine in CavAb from vertical and side view.

*Moreover, there is an opportunity for a much better discussion on DHPs that block only L-type channels versus*
*those that act on both T-types and L-types. For example, nimodipine is a good blocker of T-types, by nifedipine is*
*not. Also consider PMID: 24149495, PMID: 24990197, PMID: 30408648, PMID: 36209630, PMID: 31419643,*
*PMID: 11854446 as examples of DHP compounds that block both L-type and T-type channels – can the authors*
*come up with an explanation as to why these molecules block Cav1.2 and Cav3 channels based on their chemical*
*structures in comparison with benidipine and how it interacts with the two channel subtypes? This could be quite*
*easily done with an in silico docking program, or even by a crude comparison of these various chemical structures.*
*In my view, this would be much more useful than what is shown in Figure 2 and a much better focus with the*
*potential for much greater impact.*

**Reply:** We would like to express our sincere thanks to the reviewer for this insightful comments and
suggestions. We totally agree with that there is an opportunity for a much better discussion on DHPs that
block only L-type channels versus those that act on both L-types and T-types.

We have reviewed the literature provided and supplemented it with additional references. The L-type
selective dihydropyridines (DHPs) and L/T selective DHPs proposed in these references have been
summarized in Supplementary Table 3, and have been classified into five classes based on the chemical
structures, with corresponding structural formula listed on the left side of the table.

Classes 1, 2, and 3 have similar backbones, so we discuss them collectively. Comparing their chemical
structures and inhibitory selectivity, we find that the main differences influencing the selectivity exist in
additional modifications on the C3 and C4 benzene rings. In Class 1 and Class 2, a prominent contrast is
observed between L-type selective compounds (nifedipine, nitrendipine, felodipine, and compound 2-1)
and L/T-type selective compounds (benidipine, manidipine, compound 2-2, and compound 2-3). The C3
group in the former is smaller and relatively hydrophilic, while in the latter, it is larger and hydrophobic.

In Class 1, cilnidipine, amlodipine, and nimodipine are discussed as exceptions. Although the C3 position
of cilnidipine has a long-chain hydrophobic modification, it does not exhibit inhibitory effects on T-type
channels. We speculate that this is due to the influence of the rigid plane of alkenyl group in the C3
modification of cilnidipine. Although amlodipine shows inhibitory effects on T-type channels, its C3
modification group is smaller. We speculate that the modification group at C2 compensates for this
deficiency. The C2 amino group likely interacts with hydrophilic residues in pore of T-type channels. The
L/T-selective inhibition of nimodipine likely represents a critical threshold, indicating the minimal side
chain size required for L/T selectivity. Class 3 represents modification sites affecting inhibitory selectivity
other than the C3 group on this scaffold. L/T-selective compounds in Class 3-1 and Class 3-2 are
structurally similar to L-type selective compounds like nifedipine, nitrendipine, and felodipine, except for
longer alkyl chains on the C4 benzene ring.

Classes 4 and 5 compounds share the common feature of modification into a ring at the C5/C6 position,
so they are discussed together. Within Class 4/5 compounds, several molecules have very similar
chemical structures, so they are compared and analyzed. Compound 4-1 is L-type selective, and
compound 4-4 is L/T-type selective, with the only difference being the modification of a benzoic acid
group on the C4 benzene ring in compound 4-1, while compound 4-4 has a hydroxyl group in the

corresponding position. The structures of L-type selective compound 5-1 and L/T-type selective
compound 5-3 compounds are almost identical. The structural difference is that compound 5-1 has an
additional benzene ring on the C4 benzene ring, while compound 5-3 has a heterocyclic ring on the C4
benzene ring. Summarizing the compounds listed, for those with C5-C6 rings, whether C3 modification is
hydrophobic does not have a significant impact. The difference lies in whether there is an additional
aromatic group modification on the C4 benzene ring. The C4 group of compound 5-3 likely represent the
maximum size that T-type selectivity can tolerate.

To further investigate the interaction between DHPs with different selectivity of L-type and T-type
channels, we applied molecular docking employing Cav1.2^{BEN} as a template. We docked L-type selective
compounds from Class 1 to the L-type channel Cav1.2 and L/T-type selective compounds from Class 1
to both L-type channel Cav1.2 and T-type channel Cav3.1. Additionally, we compared the docking results
of our non-selective blockers with CaV1.2 to the resolved complex of Cav1.1 with amlodipine (7JPX)
(Gao & Yan, 2021). Regardless of L-type or L/T-type selective DHPs, they all bound in the fenestration of
D_{III}-D_{IV} in Cav1.2 or Cav3.1, with good overlap of the dihydropyridine backbone (Supplementary Fig. 9b-
c). Both L-type selective and L/T-type selective DHPs docked to Cav1.2 could form hydrogen bonds with
T1056, Q1060, and S1132. The C3 group of L-type selective DHPs docked to Cav1.2 was confined near
M1509, surrounded by a hydrophobic pocket between D_{III}S6 and D_{IV}S6. In contrast, L/T-type selective
DHPs docked to Cav3.1 could not form hydrogen bonds at the corresponding positions of these three
residues. Their C3 groups not only occupied the binding pocket formed by D_{III}S6 and D_{IV}S6 but also
extended into a hydrophobic pocket formed by D_{III}S5 and D_{IV}S5, a position that was quite similar to
where L/T-type selective DHPs docked in Cav1.2. This indicates that whether the C3 group can occupy
the binding pocket formed by D_{III}S5 and D_{IV}S5 is likely the key determinant of L-type selectivity and L/T-
type selectivity in DHPs.

Regarding the exceptional case of cilnidipine, its C3 alkene group occupied the C5 position in the Cav1.2
binding pocket. This could be due to the planar rigidity of the alkene group, which prevented it from fitting
into the C3 position. The large C5 group might clash with F1400 in Cav3.1, which might explain why,
despite cilnidipine having a larger and hydrophobic C3 group, it cannot inhibit T-type channels. As for the
other exception, amlodipine, C2 amino group in Cav3.1 extends towards the pore and forms potential
interactions with hydrophilic residues K1462 and Q1868 near the pore, supplementing the weak
interaction between C3 group and Cav3.1. L/T-type selective DHPs also exhibit varying affinities for
different T-type channels. For example, nimodipine is more inclined to inhibit Cav3.2 and Cav3.3 rather
than Cav3.1. Therefore, we aligned the Cav3.3 structure with the Cav3.1 structure. The alignment results
showed that the hydrophobic pockets in D_{III}S5 and D_{IV}S5 formed by Cav3.3 are smaller than those
formed by Cav3.1. This size difference might be a contributing factor to nimodipine's varying inhibitory
effects on different T-type channels.

Based on these analyses, a discussion was incorporated into the manuscript, starting in line 273.

[revised manuscript text omitted]

The Supplementary Fig. 9 and Supplementary Table 2 added to manuscript are attached below for your
convenience.

Figure 13* Comparison of the Interactions between Different Dihydropyridine (DHP) Drugs and Cav Channels. (Corresponding to Supplementary Fig. 9)

a. Display of L/T-type selective inhibitors benidipine in complex with Cav1.2 and amlodipine in complex with Cav1.1 (7JPX). C3 and C5 group modified on benidipine are labeled by dashed circles. **b.** Docking results of L-type selective DHP blocker in Cav1.2 are displayed. **c.** Display of docking results for L/T-type selective DHP blockers with Cav1.2. **d.** Display of docking results for L/T-selective blockers with Cav3.1 (6KZP) and Cav3.3(7WLI).

Supplementary Table 2. Summary of Selectivity of DHPs for L-type and T-type Channels

Blocker of L-type channels

Blocker of both T-type and L-type channels

*Grammar could be improved – there are numerous errors*505 **Reply:** Thanks for your kind reminder. We have carefully checked and improved the English writing in
the revised manuscript.507 *Supplemental Fig 1a is not needed – we all know what a Cav1.2 channel looks like electrophysiologically.*508 **Reply:** We appreciate the reviewer's comment. Since the construct we used to resolve the structure was

modified on the wild type, electrophysiological experiments were performed to verify that it has similar
electrophysiological properties to the wild type Cav1.2.

**REFERENCE**

- Furukawa, T., Nukada, T., Namiki, Y., Miyashita, Y., Hatsuno, K., Ueno, Y., . . . Isshiki, T. (2009). Five different profiles
of dihydropyridines in blocking T-type Ca(2+) channel subtypes (Ca(v)3.1 (alpha(1G)), Ca(v)3.2
(alpha(1H)), and Ca(v)3.3 (alpha(1I))) expressed in *Xenopus* oocytes. *European Journal of Pharmacology*,
*613*(1-3), 100-107. doi:10.1016/j.ejphar.2009.04.036
- Furukawa, T., Yamakawa, T., Midera, T., Sagawa, T., Mori, Y., & Nukada, T. (1999). Selectivities of dihydropyridine
derivatives in blocking Ca(2+) channel subtypes expressed in *Xenopus* oocytes. *J Pharmacol Exp Ther*,
*291*(2), 464-473.
- Gao, S., & Yan, N. (2021). Structural Basis of the Modulation of the Voltage-Gated Calcium Ion Channel Cav 1.1
by Dihydropyridine Compounds*. *Angewandte Chemie (International Ed. In English)*, *60*(6), 3131-3137.
doi:10.1002/anie.202011793
- He, M., Bodi, I., Mikala, G., & Schwartz, A. (1997). Motif III S5 of L-type calcium channels is involved in the
dihydropyridine binding site. A combined radioligand binding and electrophysiological study. *J Biol*
*Chem*, *272*(5), 2629-2633. doi:10.1074/jbc.272.5.2629
- McGoldrick, L. L., Singh, A. K., Saotome, K., Yelshanskaya, M. V., Twomey, E. C., Grassucci, R. A., & Sobolevsky, A. I.
(2018). Opening of the human epithelial calcium channel TRPV6. *Nature*, *553*(7687), 233-237.
doi:10.1038/nature25182
- Tang, L., Gamal El-Din, T. M., Payandeh, J., Martinez, G. Q., Heard, T. M., Scheuer, T., . . . Catterall, W. A. (2014).
Structural basis for Ca2+ selectivity of a voltage-gated calcium channel. *Nature*, *505*(7481), 56-61.
doi:10.1038/nature12775
- Tang, L., Gamal El-Din, T. M., Swanson, T. M., Pryde, D. C., Scheuer, T., Zheng, N., & Catterall, W. A. (2016).
Structural basis for inhibition of a voltage-gated Ca2+ channel by Ca2+ antagonist drugs. *Nature*,
*537*(7618), 117-121. doi:10.1038/nature19102
- Yao, X., Gao, S., & Yan, N. (2022). Structural basis for pore blockade of human voltage-gated calcium channel
Ca(v)1.3 by motion sickness drug cinnarizine. *Cell Res*, *32*(10), 946-948. doi:10.1038/s41422-022-00663-
5
- Zhao, Y., Huang, G., Wu, J., Wu, Q., Gao, S., Yan, Z., . . . Yan, N. (2019). Molecular Basis for Ligand Modulation of a
Mammalian Voltage-Gated Ca Channel. *Cell*, *177*(6). doi:10.1016/j.cell.2019.04.043

Reviewer #1 (Remarks to the Author):

The authors addressed all my previous comments and I have no further comments.

Reviewer #2 (Remarks to the Author):

Nice job on the revisions - the only thing I am wondering about are the citations in the supplemental Tables - for the channelopathies they are cited as author, year and journal, for the various DHPs they are cited as Pubmed IDs. Presumably somewhere in the manuscript and supplemental data there should be a proper reference list for all of these citations in the same format - I suspect that there is a reference limit in the main text that precludes from them being cited in the main body, but perhaps a proper reference list for all citations in the supplemental tables may need to be produced.